# PCDM: Perceptual Consistency in Diffusion Models for No-Reference Image Quality Assessment

## Abstract

Despite recent advancements in latent diffusion models that generate high-dimensional image data and perform various downstream tasks, there has been little exploration into perceptual consistency within these models on the task of No-Reference Image Quality Assessment (NR-IQA). In this paper, we hypothesize that latent diffusion models implicitly exhibit perceptually consistent local regions within the data manifold. We leverage this insight to guide on-manifold sampling using perceptual features and input measurements. Specifically, we propose **P**erceptual **M**anifold **G**uidance (PMG), an algorithm that utilizes pretrained latent diffusion models and perceptual quality metrics to obtain perceptually consistent multi-scale and multi-timestep feature maps from the denoising U-Net. We empirically demonstrate that these hyperfeatures exhibit high correlation with human perception in IQA tasks. Our method can be applied to any existing pretrained latent diffusion model and is straightforward to integrate. To the best of our knowledge, this paper is the first work to explore **P**erceptual **C**onsistency in **D**iffusion **M**odels (PCDM) and apply it to the NR-IQA problem in a zero-shot setting. Extensive experiments on IQA datasets show that our method, PCDM, achieves state-of-the-art performance, underscoring the superior zero-shot generalization capabilities of diffusion models for NR-IQA tasks. The source code will be made publicly available upon publication at `https://perceptual-consistency-in-dm.github.io`

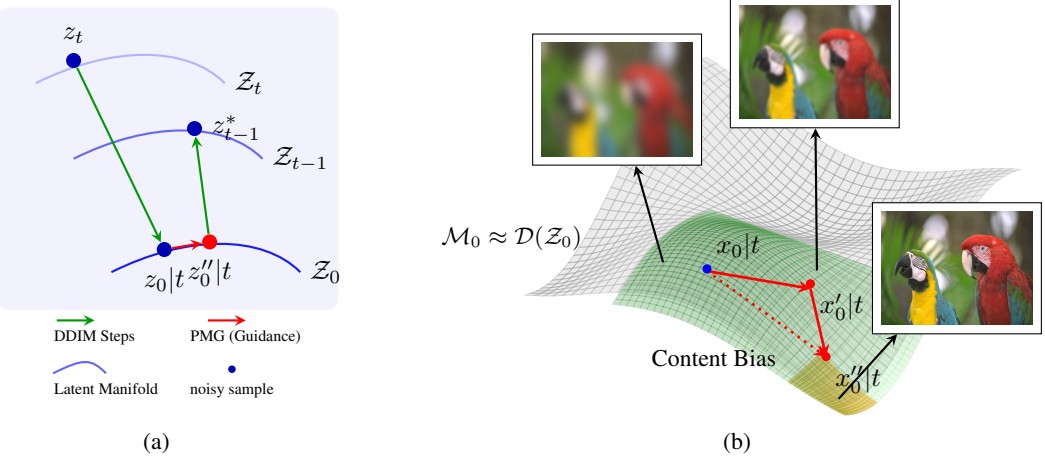

(a) (b)

Figure 1: An overview of our proposed approach: (a) shows the transition of latent samples across latent manifolds, highlighting the steps of DDIM and PMG. (b) depicts the content bias (green) on the manifold $\mathcal{M}_0 \approx \mathcal{D}(\mathcal{Z}_0)$, showing that the guidance term in red (PMG) pushes a data sample $(x'_{0|t} \sim \mathcal{D}(z'_{0|t}))$ towards the perceptually consistent region (orange) of the manifold.

## 1 INTRODUCTION

Score-based diffusion models have advanced significantly in recent years and have achieved remarkable success at synthesizing high-quality images across diverse scenes, views, and lighting conditions (Ho et al. (2020); Song & Ermon (2019); Song et al. (2020b); Zhang et al. (2023a)). Latent Diffusion Models (LDMs), which embed data into a compressed latent space, enhance computational efficiency (Rombach et al. (2022)). Diffusion models provide strong data priors that effectively capture the intricacies of high-dimensional data distributions, making them powerful for generative tasks. Conditional generation using posterior sampling has become crucial for solving various real-world low-level vision problems (Kawar et al. (2022); Chung et al. (2023); Rout et al. (2024); Song et al. (2023a)). Additionally, several methods leverage the rich internal representations of diffusion models by extracting either hand-selected single or subsets of features from a denoising U-Net for downstream tasks (Tumanyan et al. (2023); Ye et al. (2023); Xu et al. (2023a); Baranchuk et al. (2021)). Despite these advancements in addressing tasks like inverse problems, segmentation, and semantic keypoint correspondence, there has been little exploration into perceptual consistency of diffusion models for No-Reference Image Quality Assessment (NR-IQA).

NR-IQA aims to evaluate image quality in line with human perception without a high-quality reference image (Wang & Bovik (2006)). It plays a crucial role in optimizing parameters for image processing tasks, such as resizing, compression (Feng et al. (2023); Liu et al. (2023)), and enhancement (Hou et al. (2024); Fei et al. (2023); Zhang et al. (2024)). Early NR-IQA methods used hand-crafted natural scene statistics features (Zhang et al. (2015); Mittal et al. (2012); Saad et al. (2012)), and have evolved into learning-based quality metrics (Madhusudana et al. (2022); Tu et al. (2021); Ke et al. (2021); Saini et al. (2024)). While learning-based methods show promise, they often lack generalizability. With the advent of generative models, some authors have explored the use of pixel diffusion models for NR-IQA tasks (Li et al. (2024b); Babnik et al. (2024); Wang et al. (2024)). While these approaches show impressive progress, they are often *ad hoc*, focusing on tasks like quality feature denoising and image restoration by converting NR-IQA problems into Full-Reference IQA (FR-IQA) ones. Additionally, training on specific IQA datasets limits their generalizability. By contrast, our goal is to utilize pretrained latent diffusion models without fine-tuning, leveraging perceptual guidance to extract intermediate multi-scale and multi-time features, termed diffusion hyperfeatures (Luo et al. (2024)), for NR-IQA.

At the core of our method is the manifold hypothesis: real data does not occupy the entire pixel space but instead lies on a smaller underlying manifold. Previous works (Chung et al. (2022b); He et al. (2023); Sun et al. (2023)) have used the manifold concept for guided sample generation and solving inverse problems. In IQA, deep models aim to learn distortion manifolds that correlate highly with human perceptual quality (Agnolucci et al. (2024); Su et al. (2023); Guan et al. (2018); Gao et al. (2024)). These manifolds represent regions within the data manifold that contain perceptually consistent samples, with content bias further narrowing these regions. Towards further advancing progress in this direction, we propose **P**erceptual **M**anifold **G**uidance (PMG) to ensure perceptually consistent on-manifold sampling, conditioned on perceptual metric features and the quality measurement process itself. Fig. 1

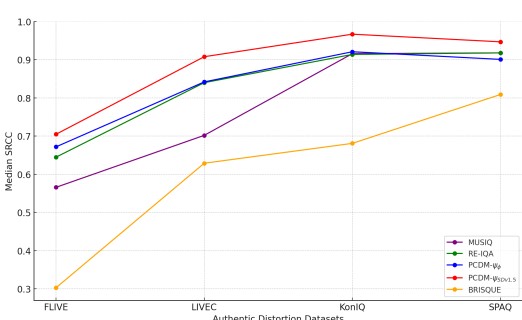

Figure 2: Median SRCC scores of NR-IQA methods across authentic distortion IQA datasets, demonstrating the superior performance of our method.

provides an overview and conceptual visualization of our approach. Unlike previous state-of-the-art CNN or transformer-based IQA models that only utilize the final feature layer, we extract intermediate multi-scale and multi-time features, termed diffusion hyperfeatures (Luo et al. (2024)), from a denoising U-Net for NR-IQA. As shown by Ghildyal et al. (2024), intermediate features of foundation models outperform state-of-the-art learned metrics based on final feature layers. Our method, **P**erceptual **C**onsistency in **D**iffusion **M**odel (PCDM), is a framework for extracting perceptually consistent diffusion hyperfeatures from unconditionally pretrained latent diffusion models,

featuring: (1) no additional fine-tuning or training, (2) generalizability across diverse distortions and image types, and (3) the first use of latent diffusion models for NR-IQA in a zero-shot setting. We theoretically prove that PMG guides sampling towards perceptually consistent regions on the image manifold in latent diffusion models, and experimental results on NR-IQA datasets support this. We evaluate PCDM against both supervised and unsupervised state-of-the-art methods on ten IQA datasets, consistently achieving superior results (see Fig. 2). With its effectiveness and generalizability, PCDM can serve as a robust framework for NR-IQA. Below, we summarize our key contributions:

- We introduce a novel approach for leveraging unconditional latent diffusion models to tackle the challenging task of No-Reference Image Quality Assessment without any fine-tuning or additional training.

- We design a manifold guidance scheme that ensures the sampling process remains on the manifold and close to perceptually consistent region. We theoretically demonstrate that our perceptual guidance keeps gradient updates on the tangent spaces of the data manifold, maintaining proximity to the local perceptually consistent manifold. We also utilize intermediate multi-scale and multi-time features from the denoising U-Net, resulting in high correlation with human perceptual judgments.

- Extensive experiments on both authentic and synthetic IQA datasets demonstrate that our method achieves state-of-the-art performance. To the best of our knowledge, this is the first approach to introduce perceptual guidance in latent diffusion models for zero-shot NR-IQA.

## 2 BACKGROUND

### 2.1 NR-IQA

Early successful NR-IQA methods relied on handcrafted features based on deviations from expected natural scene statistics (NSS) models (Ruderman & Bialek (1994); Mittal et al. (2012); Saad et al. (2012)) but these struggle to generalize across diverse and combined distortions. Deep learning introduced CNN-based models (Ke et al. (2021); Saha et al. (2023); Zhang et al. (2023b)) and transformer-based architectures such as MUSIQ (Ke et al. (2021)), TReS (Golestaneh et al. (2022b)), and TRIQ (You & Korhonen (2021)) deliver improved performance, but are limited by datasets that lacked comprehensive distortion coverage and/or inadequately large or representative numbers of human quality annotations. To better capture the complex relationship between image content and perceived quality, manifold learning techniques have been explored (Agnolucci et al. (2024); Su et al. (2023); Guan et al. (2018); Gao et al. (2024)). These approaches aim to uncover intrinsic low-dimensional structures within high-dimensional data, thereby aligning more closely with human visual perception. They generally rely on the following hypothesis:

**Assumption 1:** (Strong Manifold Hypothesis). *For a given data distribution $\mathbf{X} \in \mathbb{R}^{\mathbb{D}}$, the actual data points are concentrated on a k-dimensional locally linear subspace manifold $\mathcal{M} \subset \mathbb{R}^{\mathbb{D}}$, such that $k \ll D$ .*

Latent diffusion models have demonstrated strong representation learning capabilities when trained on large-scale datasets containing a wide range of authentic and synthetic distortions (Rombach et al. (2022); Zhang et al. (2023a)), but their application to NR-IQA problems remains underexplored (Li et al. (2024b); Babnik et al. (2024); Wang et al. (2024)).

We demonstrate that latent diffusion models implicitly learn perceptually consistent manifolds due to their extensive training data and ability to capture data priors through score matching. By leveraging the learned score function $s_\theta$ with perceptual guidance from a perceptual metric $\psi_p$, and extracting diffusion hyperfeatures $\mathbf{H} = \bigcup_{t=0}^{T} \mathbf{h_t} = \bigcup_{t=0}^{T} \bigcup_{l=0}^{L} s_\theta(x_t, t)|_l$, where $T$ represents the total sampling steps and $L$ is a subset of intermediate layers, we align features from the denoising network $s\theta$ at different time steps with human perceptual judgments.

### 2.2 DIFFUSION MODELS

**Score-Based Diffusion Models.**

We begin by reviewing the fundamentals of diffusion models, focusing on the Denoising Diffusion Probabilistic Model (DDPM) (Ho et al. (2020)). Let $x_0 \sim p(\mathbf{X})$ represent samples from the data distribution. Diffusion models define the generative process as the reverse of a noising process, which can be represented by the variance-preserving stochastic differential equation (VP-SDE) (Song et al. (2020b)) $x(t), t \in [0, T]$:

$$dx = -\frac{\beta_t}{2}xdt + \sqrt{\beta_t}d\mathbf{w} \tag{1}$$

where $\beta_t \in (0, 1)$ is the noise schedule of the process, a monotonically increasing function of $t$, and $\mathbf{w}$ is a $d$-dimensional standard Wiener process. This SDE is defined such that $x_0 \sim p(\mathbf{X})$ when $t = 0$, and as $t \to T$, the distribution approaches a standard Gaussian, i.e., $x_T \sim \mathcal{N}(\mathbf{0}, \mathbf{I})$. Our goal is to learn the reverse-time SDE corresponding to equation (1):

$$dx = \left[ -\frac{\beta_t}{2}x - \beta_t \nabla_{x_t} \log p(x_t) \right] dt + \sqrt{\beta_t}d\bar{\mathbf{w}} \tag{2}$$

where $d\bar{\mathbf{w}}$ is a reverse-time Wiener process and $dt$ runs backward, and $\nabla_{x_t} \log p(x_t)$ is the score function (Song et al., 2020b). We approximate the score function using a neural network $s_\theta(x_t, t)$ parameterized by $\theta$, trained via denoising score matching (Vincent, 2011):

$$\theta^* = \arg\min_\theta \mathbb{E}_{t \in [0,T], x_t \sim p(x_t|x_0), x_0 \sim p(\mathbf{X})} \left[ \|s_\theta(x_t, t) - \nabla_{x_t} \log p(x_t|x_0)\|_2^2 \right]. \tag{3}$$

Once $s_\theta$ is learned, we approximate the reverse-time SDE and generate clean data by iteratively solving Equation 2 from noisy samples (Song & Ermon (2019)).

**Denoising Diffusion Implicit Models (DDIM).** To address the slow generation of DDPM, Song et al. (2020a) proposed Denoising Diffusion Implicit Models (DDIMs), which define a non-Markovian diffusion process for faster sampling. The DDIM sampling update is:

$$x_{t-1} = \sqrt{\bar{\alpha}_{t-1}}\hat{x}_{0|t} + \sqrt{1 - \bar{\alpha}_{t-1} - \sigma_t^2}s_\theta(x_t, t) + \sigma_t\boldsymbol{\epsilon} \quad t = T, \ldots, 0, \tag{4}$$

where $\alpha_t = 1 - \beta_t$, $\bar{\alpha}_t = \prod_{i=1}^t \alpha_i$, $\sigma_t = \sqrt{(1 - \bar{\alpha}_{t-1})/(1 - \bar{\alpha}_t)}\sqrt{1 - \bar{\alpha}_t/\bar{\alpha}_{t-1}}$ corresponds to DDPM sampling, and when $\sigma_t = 0$ sampling becomes deterministic, where $\boldsymbol{\epsilon} \sim \mathcal{N}(\mathbf{0}, \mathbf{I})$. The term $\hat{x}_{0|t}$ is direct estimation of the clean data $x_0$ from noisy data $x_t$, calculated using Tweedie's formula (Efron (2011)):

$$\hat{x}_{0|t} = \frac{1}{\sqrt{\bar{\alpha}_t}} \left( x_t + \sqrt{1 - \bar{\alpha}_t}, s_\theta(x_t, t) \right) \tag{5}$$

**Conditional Diffusion Models.**

For conditional generation using unconditional diffusion models (Song et al. (2020b); Chung et al. (2022b); Yu et al. (2023)), a common approach is to replace the score function in equation 2 with a conditional score function $\nabla_{x_t} \log p(x_t|y)$, where $y$ is the conditioning variable. Using Bayes' rule, the conditional score function can be decomposed into the unconditional score function and a likelihood term: $\nabla_{x_t} \log p(x_t|y) = \nabla x_t \log p(x_t) + \nabla x_t \log p(y|x_t)$, Incorporating this into the reverse SDE yields:

$$dx = \left[ -\frac{\beta_t}{2}x - \beta_t \left( \nabla_{x_t} \log p(x_t) + \nabla_{x_t} \log p(y|x_t) \right) \right] dt + \sqrt{\beta_t}d\bar{\mathbf{w}} \tag{6}$$

The above SDE can be treated as a two-step process, the first getting an unconditional denoised sample $x_{t-1}$, followed by the gradient update with respect to $\mathbf{x}_t$. Since, the likelihood term $\nabla_{x_t} \log p(y|x_t)$ is generally intractable, the second term approximates a gradient update to minimizing the guidance loss around the denoised sample $x_{t-1}$.

$$x'_{t-1} = \sqrt{\bar{\alpha}_{t-1}}\hat{x}_0(x_t) + \sqrt{1 - \bar{\alpha}_{t-1} - \sigma_t^2}s_\theta(x_t, t) + \sigma_t\boldsymbol{\epsilon} \tag{7}$$

$$x_{t-1} = x'_{t-1} - \zeta\nabla x_t G(x_{0|t}, y) \tag{8}$$

where $\zeta$ is a tunable step size. Here, Tweedie's estimate $x_{0|t}$ is used since the guidance term is defined on the clean data $x_0$, i.e., $G_t(x_t, y) \approx \mathbb{E}p(x_0|x_t)[G_t(x_0, y)] \sim G(x0|t, y)$. The guidance term is optimized over a neighborhood around $x_t \in \mathbb{R}^{\mathbb{D}}$.

Many methods use equation 8 for conditional generation and various vision tasks (Chung et al. (2022b); Kawar et al. (2022); Yu et al. (2023); Song et al. (2023b)). For example, Chung et al. (2022b) define an $l_2$ loss term as $\left| y - \mathcal{A}(x_{0|t}) \right|_2^2$, where $\mathcal{A}$ represents a known differentiable forward degradation model, effectively guiding the generated sample to match the condition $y$.

## 3 PERCEPTUAL CONSISTENCY IN DIFFUSION MODEL

As discussed in Section 1, we address the limited applicability of pixel diffusion models by using the more efficient Latent Diffusion Models (LDMs). The diffusion process in LDMs naturally generates on-manifold perceptually consistent samples without requiring additional models to estimate tangent spaces of the data manifold (Srinivas et al. (2023); Bordt et al. (2023); He et al. (2023)), as we will demonstrate in this section.

In LDMs, the diffusion process operates within the latent space, training a score function $s_\theta(z_t, t)$. Let $x \in \mathbb{R}^D$ represent the original high-dimensional data, and let $\mathcal{E} : \mathbb{R}^D \to \mathbb{R}^k$ be an encoder and $\mathcal{D} : \mathbb{R}^k \to \mathbb{R}^D$ be a decoder, where $k \ll D$. The embeddings in the latent space are given by $z = \mathcal{E}(x) \in \mathbb{R}^k$.

To guide the sampling process towards the perceptually consistent region on the manifold and ensure perceptually consistent hyperfeature extraction from the denoising score function, we propose the following framework. An overview of our proposed sampling process is depicted in Fig. 1, which illustrates the step-by-step guidance for extracting perceptually aligned features in the latent space.

### 3.1 PERCEPTUAL MANIFOLD GUIDANCE

We propose using perceptual features from an input measurement $y$ derived via a perceptual metric $\psi_p$ in the conditional score function, leading to $\nabla_{z_t} \log p(z_t | \psi_p(y), y)$. The choice of $\psi_p$ is detailed in Section 4 and Appendix D. Before redefining the sampling steps, let's first consider the noisy sample manifolds.

Given Assumption 1, Chung et al. (2022a;b) show that noisy data $x_t$ is probabilistically concentrated on a $(D-1)$-dimensional manifold $\mathcal{M}_t$, which encapsulates the clean data manifold $\mathcal{M}$. Formally (see Appendix B for a detailed proof):

**Proposition 1** (Noisy Data Manifold) Let the distance function be defined as $d(x, \mathcal{M}) := \inf_{y \in \mathcal{M}} \|x - y\|_2$, and define the neighborhood around the manifold $\mathcal{M}$ as $B(\mathcal{M}; r) := \left\{ x \in \mathbb{R}^D \mid d(x, \mathcal{M}) < r \right\}$. Consider the distribution of noisy data given by $p(x_t) = \int p(x_t|x_0)p(x_0)dx_0$, $p(x_t|x_0) := \mathcal{N}(\sqrt{\bar{\alpha}_t} \mathbf{x}_0, (1 - \bar{\alpha}_t)\mathbf{I})$ represents the Gaussian perturbation of the data at time $t$, and $\bar{\alpha}_t = \prod_{s=1}^{t} \alpha_s$ is the cumulative product of the noise schedule $\alpha_t$. Under the Assumption 1, the distribution $p_t(x_t)$ is concentrated on a $(D-1)$-dim manifold $\mathcal{M}_t := y \in \mathbb{R}^D : d(y, \sqrt{\bar{x}_t}\mathcal{M}) = r_t := \sqrt{(1 - \bar{\alpha}_t)(D - k)}$.

Most posterior sampling methods (Chung et al. (2022a)) optimize the guidance term $G(x_{0|t}, y)$ over $x_t \in \mathbb{R}^D$, whereas the score function $s_\theta$ is trained only with samples on $\mathcal{M}_t$, as indicated by Proposition 1. This discrepancy implies that the solution $x_t^*$ (leading to $x_{0|t}^*$ via Tweedie's formula 5) may not reside on $\mathcal{M}_t$, resulting in a suboptimal solution (Yu et al. (2023)). To overcome this limitation, we propose a solution over $\mathcal{M}_t$. From Assumption 1, the manifold $\mathcal{M}_t$ coincides with its tangent space $\mathcal{T} x_t \mathcal{M}_t$, i.e., $\mathcal{T} x_t \mathcal{M}_t \simeq \mathbb{R}^k$ with $k \ll D$ (Park et al. (2023)). Practically, we optimize the guidance term $G(x0|t, y)$ over $x_t \in \mathcal{T} x_t \mathcal{M}_t$. This new compact solution space ensures consistent on-manifold sampling throughout the process.

The latent space of a well-trained autoencoder implicitly captures the lower-dimensional structure of the data manifold, which can be leveraged for tangent space projection (Srinivas et al. (2023); Bordt et al. (2023)). The latent processing of LDMs aids this as the samples already lie in the lower-dimensional space $\mathbb{R}^k$. Formally (proof follows He et al. (2023), see Appendix B):

**Proposition 2** (On-manifold sample with LDM) Given a perfect autoencoder, i.e. $x = \mathcal{D}(\mathcal{E}(x))$, and a gradient $\nabla_{z_{0|t}} G(z_{0|t}, y) \in \mathcal{T}_{z_0}\mathcal{Z}$ then $\mathcal{D}(\nabla_{z_{0|t}} G(z_{0|t}, y)) \in \mathcal{T}_{x_0}\mathcal{M}$.

For LDMs, the minimization of the guidance term occurs within the tangent space of the clean data manifold. This guarantees that the generated sample remains close to the real data, without deviations. Although Rout et al. (2024) do not explicitly discuss on-manifold sampling in LDMs, their results empirically suggest the inherent manifold consistency of LDMs.

Having defined consistent on-manifold sampling, we present our Perceptual Manifold Guidance (PMG). Using Bayes' theorem on our new conditional score function $\nabla_{z_t} \log p(z_t | \psi_p(y), y)$ (see

Appendix B for details):

$$\nabla_{z_t} \log p(z_t|\psi_p(y), y) \approx \nabla_{z_t} \log p(z_t) + \nabla_{z_t} \log p(\psi_p(y)|z_t) + \nabla_{z_t} \log p(y|z_t) \quad (9)$$

From (Rout et al. (2024)), LDM's intractable terms can be approximated as:

$$\nabla_{z_t} \log p(\psi_p(y)|z_t) = \nabla_{z_t} \log p(\psi_p(y)|x_{0|t} = \mathcal{D}(z_{0|t})) \quad (10)$$

$$\nabla_{z_t} \log p(y|z_t) = \nabla_{z_t} \log p(y|x_{0|t} = \mathcal{D}(z_{0|t})). \quad (11)$$

Based on Assumption 1, Propositions 1 and 2, Equations 9-11, and Lemma 2 in Appendix B, we derive the following theorem for Perceptual Manifold Guidance (proof in Appendix B):

**Theorem 1** (Perceptual Manifold Guidance) Given Assumption 1, given a perfect encoder $\mathcal{E}$, decoder $\mathcal{D}$, and an efficient score function $s_\theta(z_t, t)$, let the gradient $\nabla_{z_{0|t}} G_1(\mathcal{D}(z_{0|t}), y)$ and $\nabla_{z_{0|t}} G_2(\psi_p(\mathcal{D}(z_{0|t})), \psi_p(y))$ reside on the tangent space $\mathcal{T}_{z_{0|t}} \mathcal{Z}$ of the latent manifold $\mathcal{Z}$. Throughout the diffusion process, all update terms $z_t$ remain on noisy latent manifolds $\mathcal{Z}_t$, with $z''_{0|t}$ lying in a perceptually consistent manifold locality.

Discretized steps based on Theorem 1 can be written as:

$$z'_{0|t} \leftarrow z_{0|t} - \zeta_1 \nabla_{z_{0|t}} G_1(\mathcal{D}(z_{0|t}), y) \qquad \text{(posterior sampling step)} \quad (12)$$

$$z''_{0|t} \leftarrow z'_{0|t} - \zeta_2 \nabla_{z_{0|t}} G_2(\psi_p(\mathcal{D}(z_{0|t})), \psi_p(y)) \qquad \text{(perceptual consistency step)} \quad (13)$$

$$z^*_{t-1} \leftarrow \sqrt{\bar{\alpha}_{t-1}} z''_{0|t} - \sqrt{1 - \bar{\alpha}_{t-1} - \sigma_t^2} s_\theta(z_t, t) + \sigma_t \epsilon \qquad \text{(updated DDIM step)} \quad (14)$$

We use $G_i$ as $l_2$ functions. The perceptual consistency step in PMG (Equation 13), guides the sample to be close to the perceptual quality of the measurement. Specifically, during sampling, the perceptual guidance term adjusts the earlier estimate of the clean latent sample $z'_{0|t}$ toward a perceptually consistent locality on the tangent space of the clean latent manifold, $\mathcal{T}_{z_{0|t}} \mathcal{Z}$. From Theorem 1, all update terms $z_t$, including $z_{0|t}$, are on the manifold $\mathcal{Z}$. This ensures the sampling process remains close to a perceptually consistent region on the manifold, with $\mathcal{D}(z''_{0|t})$ closely aligned with the perceptual quality of the source measurement (see Fig. 1). We use internal representations from the denoising U-Net $s_\theta$ to measure this perceptual consistency, detailed in Section 3.2. The effectiveness of this approach is demonstrated empirically in Section 4, where the absence of the perceptual guidance term in PCDM results in suboptimal performance.

---

**Algorithm 1** PCDM: Perceptual Consistency in Diffusion Models for NR-IQA

---

**Require:** Input image $x$, encoder $\mathcal{E}(\cdot)$, decoder $\mathcal{D}(\cdot)$, score function $s_\theta(\cdot, t)$, perceptual metric $\psi_p(\cdot)$, regression model $g_\phi$, time steps $T$, guidance weights $\zeta_1, \zeta_2$
**Output:** Predicted quality score $q_p$
1: $z_0 \leftarrow \mathcal{E}(x)$                                                    //Encode input image to latent space
2: $\mathbf{H} \leftarrow \emptyset$                                           //Initialize hyperfeatures set $\mathbf{H}$ as empty
3: **for** $t = T, T-1, \ldots, 1$ **do**
4:     $\epsilon \sim \mathcal{N}(0, I)$
5:     $\epsilon_t, h_t = s_\theta(x_t, t)$                      //Estimate noise and extract feature map at time $t$
6:     $\mathbf{H} \leftarrow \mathbf{H} \cup h_t$                         //Append feature $h_t$ to hyperfeatures set $\mathbf{H}$
7:     $\hat{z}_{0|t} \leftarrow \frac{1}{\sqrt{\bar{\alpha}_t}}(z_t - \sqrt{1 - \bar{\alpha}_t} \cdot \epsilon_t)$         //Predict $\hat{z}_{0|t}$ using Tweedie's formula
8:     $z'_{0|t} \leftarrow \hat{z}_{0|t} - \zeta_1 \nabla_{z_{0|t}} G_1(\mathcal{D}(\hat{z}_{0|t}), x)$           //Posterior sampling step
9:     $z''_{0|t} \leftarrow z'_{0|t} - \zeta_2 \nabla_{z_{0|t}} G_2(\psi_p(\mathcal{D}(\hat{z}_{0|t})), \psi_p(x))$     //Perceptual consistency step
10:    $z^*_{t-1} \leftarrow \sqrt{\bar{\alpha}_{t-1}} \cdot z''_{0|t} - \sqrt{1 - \bar{\alpha}_{t-1} - \sigma_t^2} \cdot \epsilon_t + \sigma_t \epsilon$     //Update latent state
11: **end for**
12: $q_p \leftarrow g_\phi(\mathbf{H})$                                       //Predict quality score
13: **return** $q_p$

---

## 3.2 DIFFUSION HYPERFEATURES & NR-IQA

Our primary goal is to assess the perceptual quality of a given image without any reference and in a zero-shot manner. To achieve this, we propose to use *diffusion hyperfeatures*—multi-scale and multi-timestep feature maps extracted from the denoising U-Net ($s_\theta$) of a pretrained latent diffusion model.

Previous NR-IQA methods typically rely on features extracted from fine-tuned models (Ke et al. (2021); Madhusudana et al. (2022); Liu et al. (2022); Saini et al. (2024)). However, these methods often use features from the final layer or a single scale, limiting their ability to capture the complete spectrum of image characteristics. In contrast, we harness the rich hierarchical representations available in the intermediate layers of the denoising U-Net across multiple diffusion timesteps. This enables us to capture both coarse and fine-grained image features crucial for assessing perceptual quality (Ghildyal et al. (2024)). Recent studies (Xu et al. (2023a); Wu et al. (2023); Luo et al. (2024)) have shown that intermediate representations within diffusion models exhibit reliable semantic correspondences, although they have mostly been used for tasks such as data augmentation, generation, and segmentation. We hypothesize that these intermediate features also correlate strongly with human perceptual judgments of image quality, motivated by the diffusion models' ability to generate perceptually appealing images and their robust representational capabilities for various downstream tasks (Zhao et al. (2023)).

To extract these diffusion hyperfeatures, we gather intermediate feature maps from all upsampling layers of the denoising U-Net across multiple diffusion timesteps during the sampling process (see Fig. 4 in Appendix C). These feature maps inherently contain shared representations that capture different image characteristics, such as semantic content, at various scales and levels of abstraction. Since these features are distributed over both the network layers and diffusion timesteps, we aggregate $s_\theta$ layers and timesteps as diffusion hyperfeatures for NR-IQA. Specifically, the set of all extracted features is denoted as:

$$\mathbf{H} = \bigcup_{t=1}^{T} \left\{ s_\theta^{(l)}(\mathbf{x}_t) \mid l \in \mathcal{L} \right\}, \tag{15}$$

where $s_\theta^{(l)}(\mathbf{x}_t)$ represents the feature map from layer $l$ at timestep $t$, $\mathcal{L}$ is the set of layers from which we extract features, and $T$ is the total number of timesteps considered. Our experiments show that perceptual quality is built progressively during reverse diffusion (later timesteps), making an appropriate range of sampling timestep to be [0-100].

With the aggregated diffusion hyperfeatures $\mathbf{H}$, we employ a lightweight regression network $g_\phi$ parameterized by $\phi$ to predict the perceptual quality score:

$$q_p = g_\phi(\mathbf{H}), \tag{16}$$

where $q_p$ is the predicted quality score. The regression network is trained following standard NR-IQA practices (Madhusudana et al. (2022); Saha et al. (2023)), using a small dataset of images with known quality scores. Importantly, the diffusion model $s_\theta$ remains fixed and is not fine-tuned, preserving its zero-shot generalization capabilities. By leveraging perceptually rich and diverse representations, our method is better equipped to assess perceptual quality in a way that aligns with human judgments. The use of multi-scale and multi-timestep features enables the model to be sensitive to different types of distortions and image artifacts, which might not be captured when using single-scale features.

Our experiments show that PCDM with PMG provides more perceptually aligned feature maps (see Section 4). This alignment allows for a superior evaluation of image quality that reflects human perceptual judgements of visual distortions.

## 4 EXPERIMENTS

### 4.1 EXPERIMENTAL SETTINGS

To thoroughly evaluate the effectiveness of our proposed method, we conducted extensive experiments on ten publicly available and well-recognized IQA datasets, covering synthetic distortions,

| Dataset | Type | Images | Description |
|---|---|---|---|
| LIVE IQA (Sheikh et al. (2006)) | Synthetic | 779 | 29 reference images; 5 distortions at 4 levels |
| CSIQ-IQA ( Larson & Chandler (2010)) | Synthetic | 866 | 30 reference images; 6 distortions |
| TID2013 ( Ponomarenko et al. (2013)) | Synthetic | 3,000 | 25 reference images; 24 distortions at 5 levels |
| KADID-10k ( Lin et al. (2019)) | Synthetic | 10,125 | 81 reference images; 25 distortions at 5 levels |
| CLIVE ( Ghadiyaram & Bovik (2015)) | Authentic | 1,162 | Mobile images with real-world distortions |
| KonIQ-10k ( Hosu et al. (2020)) | Authentic | 10,073 | Diverse images from YFCC100M dataset |
| FLIVE ( Ying et al. (2020)) | Authentic | 39,810 | Emulates social media content |
| SPAQ ( Fang et al. (2020)) | Authentic | 11,000 | Mobile images with annotations |
| AGIQA-3K ( Li et al. (2023)) | AIGC | 3,000 | AI-generated images for IQA |
| AGIQA-1K ( Li et al. (2023)) | AIGC | 1,000 | AI-generated images for IQA |

Table 1: Summary of the IQA datasets used in our experiments.

| Methods | CLIVE | | KonIQ | | FLIVE | | SPAQ | |
|---|---|---|---|---|---|---|---|---|
| | PLCC↑ | SRCC↑ | PLCC↑ | SRCC↑ | PLCC↑ | SRCC↑ | PLCC↑ | SRCC↑ |
| ILNIQE (Zhang et al. (2015)) | 0.508 | 0.508 | 0.537 | 0.523 | - | - | 0.712 | 0.713 |
| BRISQUE (Mittal et al. (2012)) | 0.629 | 0.629 | 0.685 | 0.681 | 0.341 | 0.303 | 0.817 | 0.809 |
| WaDIQaM (Bosse et al. (2018)) | 0.671 | 0.682 | 0.807 | 0.804 | 0.467 | 0.455 | - | - |
| DBCNN (Zhang et al. (2020)) | 0.869 | 0.851 | 0.884 | 0.875 | 0.551 | 0.545 | 0.915 | 0.911 |
| TIQA (Stepien & Oszust (2023)) | 0.861 | 0.845 | 0.903 | 0.892 | 0.581 | 0.541 | - | - |
| MetaIQA (Zhu et al. (2020)) | 0.802 | 0.835 | 0.856 | 0.887 | 0.507 | 0.540 | - | - |
| P2P-BM (Ying et al. (2020)) | 0.842 | 0.844 | 0.885 | 0.872 | 0.598 | 0.526 | - | - |
| HyperIQA (Su et al. (2020)) | 0.882 | 0.859 | 0.917 | 0.906 | 0.602 | 0.544 | 0.915 | 0.911 |
| TReS (Golestaneh et al. (2022a)) | 0.877 | 0.846 | 0.928 | 0.915 | 0.625 | 0.554 | - | - |
| MUSIQ (Ke et al. (2021)) | 0.746 | 0.702 | 0.928 | 0.916 | 0.661 | 0.566 | 0.921 | 0.918 |
| RE-IQA (Saha et al. (2023)) | 0.854 | 0.840 | 0.923 | 0.914 | 0.733 | 0.645 | 0.925 | 0.918 |
| LoDA (Xu et al. (2023b)) | 0.899 | 0.876 | 0.944 | 0.932 | 0.679 | 0.578 | 0.928 | 0.925 |
| PCDM-$\psi_\phi$ | 0.853 | 0.842 | 0.929 | 0.921 | 0.751 | 0.672 | 0.912 | 0.901 |
| PCDM-$\psi_{BRISQUE}$ | 0.852 | 0.840 | 0.924 | 0.919 | 0.691 | 0.598 | 0.917 | 0.908 |
| PCDM-$\psi_{MUSIQ}$ | 0.869 | 0.858 | 0.939 | 0.928 | 0.747 | 0.672 | 0.922 | 0.920 |
| PCDM-$\zeta_1 = 0, \psi_{SDv1.5}$ | 0.901 | 0.893 | 0.952 | 0.941 | 0.799 | 0.683 | 0.931 | 0.929 |
| PCDM-$\psi_{RE-IQA}$ | 0.903 | 0.891 | 0.952 | 0.944 | 0.761 | 0.679 | 0.929 | 0.924 |
| PCDM-$\psi_{SDv1.5}$ | **0.940** | **0.908** | **0.972** | **0.967** | **0.812** | **0.705** | **0.948** | **0.947** |

Table 2: Comparison of our proposed PCDM with SOTA NR-IQA methods on PLCC and SRCC Scores for authentic IQA datasets. The best results are highlighted in bold, and the second-best results are underlined.

authentic distortions, and the latest AI-generated content (AIGC). These datasets are summarized in Table 1. Many previous methods focused only on synthetic distortions, because of the difficulty of generalizing to real-world distortions. By contrast, our LDM is pretrained on a diverse dataset that includes both synthetic and authentic distortions, allowing for a fair comparison across all types of IQA datasets, including recent AIGC datasets.

For LDM, we use the widely adopted Stable Diffusion v1.5 ( Rombach et al. (2022)), pretrained on the LAION-5B dataset ( Schuhmann et al. (2022)). We run 10 DDIM steps, with $t$ within the range $(0, 100]$ and set the hyperparameters $\zeta_1$ and $\zeta_2$ to 1 and 0.2, respectively. We discuss the implementation in detail in Appendix C. The impact of the choice of $\psi_p$ is discussed in detail in the ablation study and Appendix D. All experiments were conducted on an NVIDIA A100 GPU using PyTorch.

## 4.2 EXPERIMENTAL RESULTS & COMPARISONS

We evaluated PCDM on ten datasets. Table 2 presents the performance of PCDM on four authentic distortion ("In the Wild") datasets, with PCDM-$\psi_{SDv1.5}$ achieving the best results across all datasets. Specifically, on the CLIVE (Ghadiyaram & Bovik (2015)) dataset, PCDM-$\psi_{SDv1.5}$ attained a PLCC of 0.940 and an SRCC of 0.908, significantly surpassing the previous best method, LoDA (Xu et al. (2023b)). On the FLIVE (Ying et al. (2020)) dataset (Ying et al. (2020)), which contains the largest collection of human-labeled authentically distorted images emulating social media content (UGC), our method achieves a state-of-the-art PLCC of 0.812 and an SRCC of 0.705, demonstrating its robustness at handling diverse and complex real-world distortions. Results on synthetic distortion datasets are provided in Appendix D. We also evaluated our method on AIGC datasets to assess its ability to handle AI-generated images, which often present unique challenges.

| Method | AGIQA-1K | | AGIQA-3K | |
|---|---|---|---|---|
| | PLCC↑ | SRCC↑ | PLCC↑ | SRCC↑ |
| CONTRIQUE ( Madhusudana et al. (2022)) | 0.708 | 0.670 | 0.868 | 0.804 |
| RE-IQA ( Saha et al. (2023)) | 0.670 | 0.614 | 0.845 | 0.785 |
| GenZIQA ( De et al. (2024)) | 0.861 | 0.840 | 0.892 | 0.832 |
| PCDM-$\psi_{\text{SDv1.5}}$ | **0.903** | **0.891** | **0.929** | **0.863** |

Table 3: PLCC and SRCC comparison of PCDM on AI Generated Datasets for IQA. The best results are highlighted in bold, and the second-best results are underlined.

As shown in Table 3, PCDM-$\psi_{\text{SDv1.5}}$ outperformed previous methods on both AGIQA-1K (Li et al. (2023)) and AGIQA-3K (Li et al. (2024a)) datasets, achieving PLCC scores of 0.903 and 0.929, respectively. As compared to GenZIQA (De et al. (2024)), the previous best-performing method, our approach demonstrates significant improvements, highlighting its strong prior for AI-generated content, which is often lacking in previous methods.

### 4.3 ABLATION STUDY

We conducted cross-dataset evaluations to assess the generalization capability of our method. Table 4 presents the results of inter-dataset evaluations. Our PCDM-$\psi_{\text{SDv1.5}}$ consistently achieved the highest SRCC scores across all cross-dataset combinations, demonstrating the robustness and strong generalization capabilities of PCDM's perceptual feature maps.

We evaluated different models for $\psi_p$ to analyze their impact on perceptual guidance during the sampling process. The results, presented in Table 2, indicate that diffusion models inherently contain rich perceptual representations that provide the best guidance. IQA models like RE-IQA (Saha et al. (2023)) can still provide appropriate guidance, while models with worse human judgment correlation (e.g., $\psi_{\text{BRISQUE}}$) tend to reduce performance by deviating samples away from the perceptually consistent regions. Similar experiments were also conducted on synthetic datasets, with results available in Appendix D.

| Train | Test | Methods | | | |
|---|---|---|---|---|---|
| | | REIQA | DEIQT | LoDA | PCDM-$\psi_{\text{SDv1.5}}$ |
| FLIVE | KonIQ | 0.764 | 0.733 | 0.763 | **0.802** |
| FLIVE | CLIVE | 0.699 | 0.781 | 0.805 | **0.849** |
| KonIQ | CLIVE | 0.791 | 0.794 | 0.811 | **0.853** |
| CLIVE | KonIQ | 0.769 | 0.744 | 0.745 | **0.794** |

Table 4: SRCC Scores for Cross Dataset Evaluations. The best results are highlighted in bold, and the second-best results are underlined.

Table 7 in Appendix D shows the impact of varying the number of timesteps during sampling on FLIVE (Ying et al. (2020)) using PCDM-$\psi_{\text{SDv1.5}}$. We observe a convex trend in the SRCC scores—performance improves with an increase in the number of timesteps up to 50, but further increments result in diminishing returns, with increased computational cost. For practical use, a trade-off between accuracy and computational efficiency is required.

We also evaluated different versions of Stable Diffusion, including versions 1-3, 1-4, 1-5, 2, and 2-1. SDv1.5 achieved the best performance. Detailed results are provided in Appendix D.

We conducted experiments to evaluate the effect of different values for the hyperparameters $\zeta_1$ and $\zeta_2$. PCDM-$\psi_\phi$ in Table 2 represents the case where only the first term in PMG is used. The first term provides a strong baseline due to content bias effects from data consistency. We set $\zeta_1 = 1$ following Rout et al. (2024). In Fig. 3, we show SRCC scores for different values of $\zeta_2$ on FLIVE. We observe that values too small or large for $\zeta$ lead to poor perceptual features by pushing the samples away from the perceptually consistent region on $\mathcal{M}$.

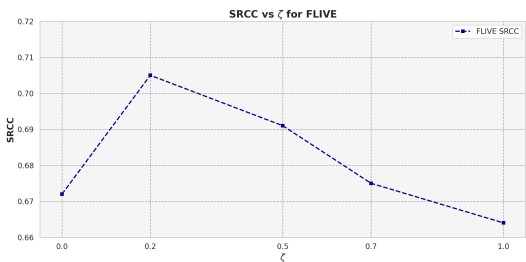

Figure 3: Effect of $\zeta_2$ on the hyperfeatures, realised on SRCC.

It may be observed that different layers contribute differently to the overall performance.

In particular, Layer 4 contributes significantly
more to final quality prediction than other layers. Details can be found in Appendix D.

## 5 CONCLUSION

In this work, we introduced the Perceptually Consistent Diffusion Model (PCDM) for No-Reference Image Quality Assessment (NR-IQA). Leveraging the strong representation capabilities of pretrained latent diffusion models (LDMs), we proposed Perceptual Manifold Guidance (PMG) to direct the sampling process toward perceptually consistent regions on the data manifold. We demonstrated the value of extracting multi-scale and multi-timestep features—diffusion hyperfeatures from the denoising U-Net, providing a rich representation for quality assessment. To our knowledge, this is the first work to utilize pretrained LDMs for NR-IQA in a zero-shot manner.

## 6 REPRODUCIBILITY STATEMENT

Reproducibility is a key aspect of our contribution. Upon publication, we will provide a public codebase to facilitate the replication of our experiments. All datasets used are publicly available, and details on dataset splits for training, validation, and testing are provided. Supplementary materials include theoretical proofs and ablation studies on different hyperparameters and configurations to support reproducibility.

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

# Appendix

Here, we provide additional theoretical proof, implementation details, and experimental results to complement those in the main paper. Specifically, Section A discusses more related work and background on diffusion models and NR-IQA, Section B presents detailed theoretical proofs and supporting discussionn, Section C describes the implementation details, Section D includes further quantitative analyses to demonstrate the performance of PCDM, Finally in Section E we discuss the main limitation of our proposed method and possible extensions.

## A    RELATED WORK

### A.1    NR-IQA

No-Reference Image Quality Assessment (NR-IQA) has been a focal point of research over the past two decades, aiming to evaluate image quality based on human perception without relying on reference images. Early approaches predominantly utilized handcrafted features derived from natural scene statistics (NSS), with models such as BRISQUE (Mittal et al. (2012), DIIVINE Moorthy & Bovik (2011), BLIINDS Saad et al. (2012), and NIQE Mittal et al. (2012)). While these methods effectively leveraged statistical regularities in natural images, their performance often suffered when dealing with complex or unseen distortions due to their reliance on specific statistical models.

The emergence of deep learning introduced convolutional neural networks (CNNs) into NR-IQA, enabling models to learn hierarchical feature representations directly from data. Transformer-based architectures further advanced the field by capturing long-range dependencies and contextual information, with models such as MUSIQ (Ke et al. (2021), TReS Golestaneh et al. (2022b), and TRIQ You & Korhonen (2021)) demonstrating significant improvements in performance. Despite these advancements, a major limitation persists: the lack of large-scale, diverse datasets encompassing the full spectrum of real-world distortions. This scarcity hampers the generalization capabilities of NR-IQA models, as they are trained on datasets that do not adequately represent all possible image degradation scenarios.

To tackle the complexity of image distortions, the concept of perceptual or distortion manifolds has been explored in image quality assessment models. Manifold learning techniques aim to uncover the intrinsic low-dimensional structures within high-dimensional data, which better align with human visual perception. For instance, Jiang et al. Jiang et al. (2018) applied manifold learning to reduce the dimensionality of RGB images, constructing low-dimensional representations for stereoscopic image quality assessment. Similarly, Guan et al. Guan et al. (2017) employed manifold learning on feature maps to capture the intrinsic geometric structures of high-dimensional data in a low-dimensional space, thereby enhancing prediction accuracy for High-Dynamic-Range (HDR) images. These approaches highlight the potential of manifold learning in modeling the complex relationships between image content and perceived quality.

Although diffusion models (DMs) have demonstrated remarkable efficacy in generating high-dimensional data and capturing rich feature representations within their intermediate layers (Ho et al. (2020); Song et al. (2020a)), their application to NR-IQA has been minimal. Existing works incorporating DMs often use them for specific tasks, such as quality feature denoising or image restoration, effectively converting NR-IQA into full-reference IQA (FR-IQA) problems (Li et al. (2024b); Babnik et al. (2024)). Typically, these methods involve training on specific IQA datasets, limiting their generalizability to diverse distortions.

In our work, we demonstrate that since diffusion models are trained on large-scale datasets containing user-generated content (UGC) images—with a wide range of authentic and synthetic distortions—they inherently learn perceptually consistent manifolds. Although these models are not specifically trained for IQA tasks, they are designed to capture data priors by learning score functions, enabling them to model complex data distributions and capture both high-level and low-level features. This capability allows them to generate a diverse set of images with fine details. We believe

that, with appropriate perceptual guidance, it is possible to extract features from diffusion models that correlate highly with human perception, in a zero-shot setting.

Diffusion models have also demonstrated the ability to learn meaningful representations within their U-Net architectures, as evidenced by studies that leverage intermediate features for various vision tasks (Zhao et al. (2023); Wu et al. (2023)). This suggests an opportunity to harness these models for NR-IQA, which has so far remained underexplored. Our work aims to address this gap by utilizing pretrained diffusion models without any fine-tuning, thereby preserving their inherent generalization capabilities. By extracting multi-scale and multi-time-step features—referred to as diffusion hyperfeatures—and incorporating perceptual guidance, we propose a method that overcomes the limitations of current NR-IQA approaches. This strategy leverages the rich representations within diffusion models to improve generalization across diverse image distortions, aligning more closely with human perceptual judgments.

## A.2 DIFFUSION MODELS

Diffusion models consist of a forward noise process and a backward denoising process. In the discrete formulation Song et al. (2020b); Ho et al. (2020), the forward process manifests as a Markov chain described by:

$$q(\mathbf{x}_{1:N} \mid \mathbf{x}_0) = \prod_{k=1}^{N} q(\mathbf{x}_k \mid \mathbf{x}_{k-1}), \quad q(\mathbf{x}_k \mid \mathbf{x}_{k-1}) = \mathcal{N}(A_k \mathbf{x}_{k-1}, b_k^2 I). \tag{A-1}$$

The coefficients $\{a_k\}_{k=1}^N$ and $\{b_k\}_{k=1}^N$ are manually set and may differ depending on various diffusion formulations Song et al. (2020b). Given that each Markov step $q(\mathbf{x}_k \mid \mathbf{x}_{k-1})$ is a linear Gaussian model, the resultant marginal distribution $q(\mathbf{x}_k \mid \mathbf{x}_0)$ assumes a Gaussian form, $\mathcal{N}(c_k \mathbf{x}_0, d_k^2 I)$. The parameters $\{c_k\}_{k=1}^N$ and $\{d_k\}_{k=1}^N$ can be derived from $\{a_k\}_{k=1}^N$ and $\{b_k\}_{k=1}^N$. For sample generation, we train a neural network, $s_\theta(\mathbf{x}_k, t_k)$, to estimate the score function $\nabla_{\mathbf{x}_k} \log q(\mathbf{x}_k \mid \mathbf{x}_0)$. The backward process, which we assume to be a Markov chain, is typically represented as:

$$p_\theta(\mathbf{x}_{k-1} \mid \mathbf{x}_k) = \mathcal{N}(u_k \hat{\mathbf{x}}_0(\mathbf{x}_k) + v_k s_\theta(\mathbf{x}_k, t_k), w_k^2 I) \tag{A-2}$$

where $\hat{\mathbf{x}}_0(\mathbf{x}_k) := \mathbf{x}_k + d_k^2 s_\theta(\mathbf{x}_k, t_k)/c_k$ is the predicted $\mathbf{x}_0$ obtained from the Tweedie's formula. Here $\{u_k\}_{k=1}^N$, $\{v_k\}_{k=1}^N$, and $\{w_k\}_{k=1}^N$ can be computed from the forward process coefficients $\{a_k\}_{k=1}^N$ and $\{b_k\}_{k=1}^N$. The formulation in Equation A-2 encompasses many stochastic samplers of diffusion models, including the ancestral sampler in DDPM (Ho et al. (2020)), and the DDIM sampler in Song et al. (2020b). For variance-preserving diffusion models Ho et al. (2020), we have:

$$a_k = \sqrt{\alpha_k}, \quad b_k = \sqrt{\beta_k}, \quad c_k = \sqrt{\bar{\alpha}_k}, \quad d_k = \sqrt{1 - \bar{\alpha}_k}, \tag{A-3}$$

where $\alpha_k := 1 - \beta_k$, $\bar{\alpha}_k := \prod_{j=1}^{k} \alpha_j$, and $\alpha_k, \beta_k$ follow the notations in Ho et al. (2020). DDPM sampling:

$$u_k = \sqrt{\alpha_{k-1}}, \quad v_k = -\sqrt{\alpha_k}(1 - \bar{\alpha}_{k-1}), \quad w_k = \sqrt{\beta_k} \cdot \sqrt{\frac{1 - \bar{\alpha}_{k-1}}{1 - \bar{\alpha}_k}}, \tag{A-4}$$

and for DDIM sampling Song et al. (2020b), we have:

$$u_k = \sqrt{\alpha_k}, \quad v_k = \sqrt{1 - \bar{\alpha}_{k-1} - \sigma_k^2} \cdot \sqrt{1 - \bar{\alpha}_k}, \quad w_k = \sigma_k, \tag{A-5}$$

where the conditional variance sequence $\{\sigma_k\}_{k=1}^N$ can be arbitrary. And depedning on the value of $\sigma_k^2$, it can become DDPM or DDIM sampling, i.e. With $\beta_k \cdot \frac{(1 - \bar{\alpha}_{k-1})}{(1 - \bar{\alpha}_k)}$ it become DDPM.

## B THEORETICAL PROOFS

### B.1 LEMMA 1 (TWEEDIE'S FORMULA FOR EXPONENTIAL FAMILY)

Let $p(z|\eta)$ belong to the exponential family distribution:

$$p(z \mid \eta) = p_0(z) \exp\left(\eta^\top T(z) - \Phi(\eta)\right) \tag{B-1}$$

where $\eta$ is the natural or canonical parameter of the family, $\Phi(\eta)$ is the cumulant generating function (cfg) (which makes $p_\eta(z)$ integrate to 1), and $p_0(z)$ is the density when $\eta = 0$. Then, the posterior mean $\hat{\eta} := \mathbb{E}[\eta \mid z]$ should satisfy:

$$(\nabla_z T(z))^\top \hat{\eta} = \nabla_z \log p(z) - \nabla_z \log p_0(z) \tag{B-2}$$

**Proof.** The marginal distribution $p(z)$ can be expressed as:

$$p(z) = \int_{\mathcal{Z}} p_\eta(z) p(\eta) d\eta \tag{B-3}$$

which, using the form of $p_\eta(z)$, becomes:

$$p(z) = p_0(z) \int_{\mathcal{Z}} \exp\left(\eta^\top T(z) - \Phi(\eta)\right) p(\eta) d\eta \tag{B-4}$$

Taking the derivative of $p(z)$ with respect to $z$:

$$\nabla_z p(z) = \nabla_z p_0(z) \int_{\mathcal{Z}} \exp\left(\eta^\top T(z) - \Phi(\eta)\right) p(\eta) d\eta + \\ \int_{\mathcal{Z}} (\nabla_z T(z))^\top \eta p_0(z) \exp\left(\eta^\top T(z) - \Phi(\eta)\right) p(\eta) d\eta \tag{B-5}$$

Rearranging, we get:

$$\nabla_z p(z) = \frac{\nabla_z p_0(z)}{p_0(z)} p(z) + (\nabla_z T(z))^\top \int_{\mathcal{Z}} \eta p_\eta(z) p(\eta) d\eta \tag{B-6}$$

which simplifies to:

$$\nabla_z p(z) = \frac{\nabla_z p_0(z)}{p_0(z)} p(z) + (\nabla_z T(z))^\top \int_{\mathcal{Z}} \eta p_z(\eta) d\eta. \tag{B-7}$$

Thus:

$$\frac{\nabla_z p(z)}{p(z)} = \frac{\nabla_z p_0(z)}{p_0(z)} + (\nabla_z T(z))^\top \mathbb{E}[\eta \mid z] \tag{B-8}$$

Finally:

$$(\nabla_z T(z))^\top \mathbb{E}[\eta \mid z] = \nabla_z \log p(z) - \nabla_z \log p_0(z). \tag{B-9}$$

This concludes the proof.

### B.2 PROPOSITION 3 (TWEEDIE'S FORMULA FOR SDE)

For the case of VP-SDE, we can estimate $p(z_0|z_t)$ as:

$$z_{0|t} := \mathbb{E}[z_0 \mid z_t] = \frac{1}{\sqrt{\bar{\alpha}(t)}} \left(z_t + (1 - \bar{\alpha}(t)) \nabla_{z_t} \log p_t(z_t)\right) \tag{B-10}$$

**Proof.** For the case of VP-SDE, we have

$$p(z_t|z_0) = \frac{1}{(2\pi(1 - \bar{\alpha}(t)))^{d/2}} \exp\left(-\frac{\|z_t - \sqrt{\bar{\alpha}(t)} z_0\|^2}{2(1 - \bar{\alpha}(t))}\right) \tag{B-11}$$

A Gaussian distribution. We can get the canonical decomposition as:

$$p(z_t|z_0) = p_0(z_t) \exp\left(z_0^\top T(z_t) - \Phi(z_0)\right), \tag{B-12}$$

And,

$$p_0(z_t) := \frac{1}{(2\pi(1 - \bar{\alpha}(t)))^{d/2}} \exp\left(-\frac{\|z_t\|^2}{2(1 - \bar{\alpha}(t))}\right) \tag{B-13}$$

$$T(z_t) := \frac{\sqrt{\bar{\alpha}(t)}}{1 - \bar{\alpha}(t)} z_t \tag{B-14}$$

$$\Phi(z_0) := \frac{\bar{\alpha}(t)\|z_0\|^2}{2(1 - \bar{\alpha}(t))} \tag{B-15}$$

Therefore, from **Lemma 1**:

$$\frac{\sqrt{\bar{\alpha}(t)}}{1 - \bar{\alpha}(t)} \hat{z}_0 = \nabla_{z_t} \log p_t(z_t) + \frac{1}{1 - \bar{\alpha}(t)} z_t \tag{B-16}$$

Giving us:

$$z_{0|t} = \frac{1}{\sqrt{\bar{\alpha}(t)}} \left(z_t + (1 - \bar{\alpha}(t))\nabla_{z_t} \log p_t(z_t)\right) \tag{B-17}$$

This concludes the proof.

### B.3 CONDITIONAL SCORE FUNCTIONS

As mentioned in the main paper, conditional score function can be written as (Equation 9 ):

$$
\begin{aligned}
\nabla_{z_t} \log p(z_t|\psi_p(y), y) \approx{} & \nabla_{z_t} \log p(z_t) \\
& + \nabla_{z_t} \log p(y|x_{0|t} = \mathcal{D}(z_{0|t})) \\
& + \nabla_{z_t} \log p(\psi_p(y)|x_{0|t} = \mathcal{D}(z_{0|t}))
\end{aligned} \tag{B-18}
$$

Proof. From Baye's theorem we can write the conditional distribution as:

$$p(z|\psi(y), y) = p(\psi(y)|z_t)p(y|z_t, \psi(y))p(z_t) \tag{B-19}$$

Note, $y$ is conditionally independent of $\psi(y)$ given $z_t$ for later timesteps in diffusion process as $z_t$ gives more structural information for image. Therefore:

$$p(z|\psi(y), y) = p(\psi(y)|z_t)p(y|z_t)p(z_t) \tag{B-20}$$

Our score function becomes:

$$\nabla_{z_t} \log p(z_t|\psi_p(y), y) \approx \nabla_{z_t} \log p(z_t) + \nabla_{z_t} \log p(\psi_p(y)|z_t) + \nabla_{z_t} \log p(y|z_t) \tag{B-21}$$

We can write the posterior as:

$$p(y|z_t) = \int p(y|z_0)p(z_0|z_t)dz_0 \tag{B-22}$$

Following (Chung et al. (2022a)) and Proposition 3, we can have the posterior as:

$$p(y|z_t) \approx p(y|z_{0|t}) \tag{B-23}$$

Therefore:

$$\nabla_{z_t} \log p(z_t|\psi_p(y), y) \approx \nabla_{z_t} \log p(z_t) + \nabla_{z_t} \log p(\psi_p(y)|z_{0|t}) + \nabla_{z_t} \log p(y|z_{0|t}) \tag{B-24}$$

Following (Rout et al. (2024)), we can approximately write the conditional probability for LDM given decoder $\mathcal{D}$:

$$p(y|z_t) \approx p(y|x_0 = \mathcal{D}(z_{0|t})) \tag{B-25}$$

Note that we ignore the gluing term proposed by (Rout et al. (2024)) as it depends on the forward degradation model only valid for inverse problems. Our final conditional score function becomes:

$$\nabla_{z_t} \log p(z_t|\psi_p(y), y) \approx \nabla_{z_t} \log p(z_t) + \nabla_{z_t} \log p(y|x_{0|t} = \mathcal{D}(z_{0|t})) + \nabla_{z_t} \log p(\psi_p(y)|x_{0|t} = \mathcal{D}(z_{0|t})) \tag{B-26}$$

### B.4 PROPOSITION 1 (NOISY DATA MANIFOLD)

Let the distance function be defined as $d(x, \mathcal{M}) := \inf_{y \in \mathcal{M}} \|x - y\|_2$, and define the neighborhood around the manifold $\mathcal{M}$ as $B(\mathcal{M}; r) := \left\{x \in \mathbb{R}^D \mid d(x, \mathcal{M}) < r\right\}$. Consider the distribution of noisy data given by $p(x_t) = \int p(x_t|x_0)p(x_0)dx_0$, $p(x_t|x_0) := \mathcal{N}(\sqrt{\bar{\alpha}_t}\,\mathbf{x}_0, \ (1 - \bar{\alpha}_t)\mathbf{I})$ represents the Gaussian perturbation of the data at time $t$, and $\bar{\alpha}_t = \prod_{s=1}^{t} \alpha_s$ is the cumulative product of the

noise schedule $\alpha_t$. Under the Assumption 1, the distribution $p_t(x_t)$ is concentrated on a $(D-1)$-dim manifold $\mathcal{M}_t := y \in \mathbb{R}^D : d(y, \sqrt{\bar{x}_t}\mathcal{M}) = r_t := \sqrt{(1-\bar{\alpha}_t)(D-k)}$.

Proof. (Mainly follow Chung et al. (2022b)):

We begin by defining the manifold $\mathcal{M}$ as $\mathcal{M} := \left\{ x \in \mathbb{R}^D : x_{k+1:D} = 0 \right\}$ which represents a subspace where the last $D - k$ coordinates are zero. Essentially, this means that $\mathcal{M}$ lies within a lower-dimensional subspace of $\mathbb{R}^D$. Let $X$ be a $\chi^2$ random variable with $n$ degrees of freedom. We use the following concentration bounds:

$$\mathbb{P}(X - n \geq 2\sqrt{n\tau} + 2\tau) \leq e^{-\tau}, \tag{B-27}$$

$$\mathbb{P}(X - n \leq -2\sqrt{n\tau}) \leq e^{-\tau}. \tag{B-28}$$

Now, consider the quantity $\sum_{i=k+1}^{D} \frac{x_{t,i}^2}{1-\bar{\alpha}_t}$, which follows a $\chi^2$ distribution with $D - k$ degrees of freedom. Using the concentration bounds and setting $\tau = (D - k)\epsilon'$, we can express the following bound:

$$\mathbb{P}\left( -2(D-k)\sqrt{\epsilon'} \leq \sum_{i=k+1}^{D} \frac{x_{t,i}^2}{1-\bar{\alpha}_t} - (D-k) \leq 2(D-k)(\sqrt{\epsilon'} + \epsilon') \right) \geq 1 - \delta. \tag{B-29}$$

The above inequality gives us a range for the summation of the squared components of $x_t$ beyond the first $k$ dimensions. We can now rewrite this in terms of the Euclidean norm of these components:

$$\mathbb{P}\left( \sqrt{\sum_{i=k+1}^{D} x_{t,i}^2} \in \left( r_t\sqrt{\max\{0, 1-2\sqrt{\epsilon'}\}}, r_t\sqrt{1+2\sqrt{\epsilon'}+2\epsilon'} \right) \right) \geq 1 - \delta, \tag{B-30}$$

where we have defined:

$$r_t := \sqrt{(1-\bar{\alpha}_t)(D-k)}. \tag{B-31}$$

To ensure that the probability holds for a given confidence level $1 - \delta$, we define:

$$\epsilon'_{t,D-k} = -\frac{1}{D-k} \log \frac{\delta}{2}. \tag{B-32}$$

We then use $\epsilon'_{t,D-k}$ to define:

$$\epsilon_{t,D-k} = \min \left\{ 1, \sqrt{\max\{0, 1-2\sqrt{\epsilon'_{t,D-k}}\}} + \frac{1 + 2\sqrt{\epsilon'_{t,D-k}} + 2\epsilon'_{t,D-k} - 1}{\sqrt{1-\bar{\alpha}_t}(D-k)} \right\}, \tag{B-33}$$

which ensures $0 < \epsilon_{t,D-k} \leq 1$. This value $\epsilon_{t,D-k}$ helps in determining the size of the neighborhood around the manifold $\mathcal{M}_t$, such that:

$$\mathbb{P}\left( x_t \in B(\mathcal{M}_t; \epsilon_{t,D-k} \cdot \sqrt{(1-\bar{\alpha}_t)(D-k)}) \right) \geq 1 - \delta. \tag{B-34}$$

Thus, we have shown that the noisy data distribution $p(x_t)$ is concentrated within a certain neighborhood around the manifold $\mathcal{M}_t$, with high probability. The parameter $\epsilon_{t,D-k}$ is decreasing with respect to $\delta$ and $D - k$, because $\epsilon'_{t,D-k}$ is also decreasing in these parameters, and $\epsilon_{t,D-k}$ is an increasing function of $\epsilon'_{t,D-k}$.

This concludes the proof.

In pixel space, as discussed by Chung et al. (2022a), the optimization of the guidance term occurs in the entire space $\mathbb{R}^D$. However, from Proposition 1, we know that $x_t$ actually lies in a much smaller subspace of $\mathbb{R}^D$, specifically in $\mathbb{R}^k$. To prevent sampling from deviating from the content-bias region on the manifold, one obvious way to improve the sampling process is to restrict the optimization space to $M_t$, specifically to the tangent space $\mathcal{T}xt\mathcal{M}t$. Previous literature has suggested using autoencoders to approximate this tangent space $\mathcal{T}x_t\mathcal{M}_t$ (Shao et al. (2018)). However, since autoencoders are not trained on intermediate noisy samples, their practical effectiveness is limited. We instead use Latent Diffusion Models (LDM), where the entire sampling process occurs in the latent space $\mathcal{M}$. This approach ensures overall data consistency, but it may still not fully achieve perceptual consistency within the content-bias region on the manifold $\mathcal{M}$ (see Fig. 1 for an intuitive illustration).

## B.5 PROPOSITION 2 (ON-MANIFOLD SAMPLE WITH LDM)

Given a perfect autoencoder, i.e. $x = \mathcal{D}(\mathcal{E}(x))$, and a gradient $\nabla_{z_{0|t}} G(z_{0|t}, y) \in \mathcal{T}_{z_0}\mathcal{Z}$ then $\mathcal{D}(\nabla_{z_{0|t}} G(z_{0|t}, y)) \in \mathcal{T}_{x_0}\mathcal{M}$.

**Proof.** We begin by considering a perfect autoencoder, consisting of an encoder $\mathcal{E}$ and a decoder $\mathcal{D}$, which satisfies the property $x = \mathcal{D}(\mathcal{E}(x))$. for any data point $x \in X \subset \mathcal{M}$. Let $z_0 = \mathcal{E}(x_0)$ be the latent representation of $x_0$. Since the autoencoder is perfect, we have $x_0 = \mathcal{D}(z_0)$.

To understand how the encoder and decoder interact in terms of their mappings, we consider their Jacobians. The Jacobian of the encoder $\frac{\partial \mathcal{E}}{\partial x_0}$ maps changes in the data space $\mathbb{R}^D$ to changes in the latent space $\mathbb{R}^k$. The Jacobian of the decoder $\frac{\partial \mathcal{D}}{\partial z_0}$ maps changes in the latent space $\mathbb{R}^k$ back to the data space $\mathbb{R}^D$. Since the autoencoder is perfect, encoding and then decoding must recover the original input exactly. This implies that the composition of the encoder and decoder Jacobians must yield the identity mapping:

$$\frac{\partial \mathcal{E}}{\partial x_0} \frac{\partial \mathcal{D}}{\partial z_0} = I, \tag{B-35}$$

where $I$ is the identity matrix. This property ensures that the encoder and decoder are exact inverses of each other in terms of their linear mappings at $x_0$ and $z_0$.

Consider a gradient $\nabla_{z_{0|t}} G(z_{0|t}, y) \in \mathcal{T}_{z_0}\mathcal{Z}$, where $\mathcal{T}_{z_0}\mathcal{Z}$ is the tangent space of the latent space $\mathcal{Z}$ at $z_0$. We want to determine the behavior of this gradient when mapped back to the data space using the decoder. The decoder Jacobian $\frac{\partial \mathcal{D}}{\partial z_0}$ maps vectors from the latent space to the data space. Since $\nabla_{z_{0|t}} G(z_{0|t}, y)$ is in the tangent space $\mathcal{T}_{z_0}\mathcal{Z}$, applying the decoder Jacobian gives:

$$\mathcal{D}(\nabla_{z_{0|t}} G(z_{0|t}, y)) = \frac{\partial \mathcal{D}}{\partial z_0} \nabla_{z_{0|t}} G(z_{0|t}, y). \tag{B-36}$$

Since the Jacobian $\frac{\partial \mathcal{D}}{\partial z_0}$ maps changes in the latent space to corresponding changes in the data space, and the latent space $\mathcal{Z}$ is designed to represent the underlying data manifold $\mathcal{M}$, it follows that:

$$\frac{\partial \mathcal{D}}{\partial z_0} : \mathcal{T}_{z_0}\mathcal{Z} \to \mathcal{T}_{x_0}\mathcal{M}. \tag{B-37}$$

Thus, the vector $\mathcal{D}(\nabla_{z_{0|t}} G(z_{0|t}, y))$ lies in the tangent space $\mathcal{T}_{x_0}\mathcal{M}$ of the data manifold at $x_0$. This implies that the gradient update, when mapped back to the data space, remains on the data manifold, ensuring consistency in the sampling process.

This concludes the proof.

## B.6 LEMMA 2 (DISTRIBUTION CONCENTRATION)

Consider the optimality of the diffusion model, i.e., $\epsilon_\theta \left( \sqrt{\alpha_t} z + \sqrt{1 - \alpha_t} \epsilon_t, t \right) = \epsilon_t$ for $z \in \mathcal{Z}$. For some $\epsilon \sim \mathcal{N}(0, I)$, the sum of noise components $\sqrt{1 - \bar{\alpha}_{t-1} - \sigma_t^2} \epsilon_\theta(z_t, t) + \sigma_t \epsilon_t$ in DDIM sampling can be expressed as:

$$\sqrt{1 - \bar{\alpha}_{t-1} - \sigma_t^2} \epsilon_\theta(z_t, t) + \sigma_t \epsilon_t = \sqrt{1 - \bar{\alpha}_{t-1}} \tilde{\epsilon}, \tag{B-38}$$

where $\tilde{\epsilon} \sim \mathcal{N}(0, I)$. Since $\sqrt{1 - \bar{\alpha}_{t-1} - \sigma_t^2} \epsilon_\theta(z_t, t)$ and $\sigma_t \epsilon_t$ are independent, their sum is also a Gaussian random variable with a mean of 0 and a variance of $(1 - \bar{\alpha}_{t-1} - \sigma_t^2) + \sigma_t^2 = (1 - \bar{\alpha}_{t-1})$.

Furthermore, let the latent data distribution $p(z)$ be a probability distribution with support on the linear manifold $\mathcal{M}$ that satisfies Assumption 1. For any $z \sim p(z)$, consider

$$z_{t-1} = \sqrt{\bar{\alpha}_{t-1}} z + \sqrt{1 - \bar{\alpha}_{t-1} - \sigma_t^2} \epsilon_\theta(z_t, t) + \sigma_t \epsilon_t. \tag{B-39}$$

Then, the marginal distribution $\hat{p}_{t-1}(z_{t-1})$, which is defined as:

$$\hat{p}_{t-1}(z_{t-1}) = \int \mathcal{N}\left(z_{t-1}; \sqrt{\bar{\alpha}_{t-1}}z + \sqrt{1 - \bar{\alpha}_{t-1} - \sigma_t^2}\epsilon_\theta(z_t, t), \sigma_t^2 I\right) p(z_t|z)p(z)\,dz\,dz_t,$$

$$(\text{B-40})$$

is probabilistically concentrated on $\mathcal{Z}_{t-1}$ for $\epsilon_t \sim \mathcal{N}(0, I)$.

**Proof.** Since $\epsilon_\theta(z_t, t)$ is independent of $\epsilon_t$, their sum is the sum of independent Gaussian random variables, resulting in a Gaussian distribution with a variance $(1 - \bar{\alpha}_{t-1})$. By this result, the multivariate normal distribution has a mean $\sqrt{\bar{\alpha}_{t-1}}z$ and a covariance matrix $(1 - \bar{\alpha}_{t-1})I$. Consequently, the marginal distribution of the target can be represented as:

$$\hat{p}_{t-1}(z_{t-1}) = \int \mathcal{N}\left(z_{t-1}; \sqrt{\bar{\alpha}_{t-1}}z, (1 - \bar{\alpha}_{t-1})I\right) p(z)\,dz,$$

$$(\text{B-41})$$

which matches the marginal distribution defined in Proposition 1. Therefore, in accordance with Proposition 1, the probability distribution $\hat{p}_{t-1}(z_{t-1})$ probabilistically concentrates on $\mathcal{M}_{t-1}$. $\square$

### B.7 THEOREM 1 (PERCEPTUAL MANIFOLD GUIDANCE)

Given Assumption 1, for perfect encoder $\mathcal{E}$, decoder $\mathcal{D}$, and an efficient score function $s_\theta(z_t, t)$, let gradient $\nabla_{z_{0|t}} G_1(\mathcal{D}(z_{0|t}), y)$ and $\nabla_{z_{0|t}} G_2(\psi_p(\mathcal{D}(z_{0|t})), \psi_p(y))$ reside on the tangent space $\mathcal{T}_{z_{0|t}}\mathcal{Z}$ of latent manifold $\mathcal{Z}$. Throughout the diffusion process, all update terms $z_t$ remain on noisy latent manifolds $\mathcal{Z}_t$, with $z_{0|t}''$ in perceptually consistent manifold locality.

**Proof.** We begin by establishing that both the gradients for data and perceptual consistency are constrained to the tangent space of the latent manifold, ensuring that updates remain on the manifold during the diffusion process. At $t = T$, we consider the noisy sample $z_T$ generated from a Gaussian distribution. Noisy sample is expressed as:

$$z_T = \sqrt{\bar{\alpha}_T}z_0 + \sqrt{1 - \bar{\alpha}_T}\epsilon_T, \quad \epsilon_T \sim \mathcal{N}(0, I) \tag{B-42}$$

where $z_0 = \mathcal{E}(x_0)$ represents the latent variable corresponding to the clean sample $x_0$. The support of the distribution $p(z_0)$ lies on the manifold $\mathcal{Z}$, ensuring that $z_0 \in \mathcal{Z}$. Assume that for all $t \geq T_1$, there exists a $z_0 \in \mathcal{Z}$ such that:

$$z_t = \sqrt{\bar{\alpha}_t}z_0 + \sqrt{1 - \bar{\alpha}_t}\epsilon, \quad \epsilon \sim \mathcal{N}(0, I) \tag{B-43}$$

We aim to prove that this also holds for $t = T_1 - 1$. At timestep $t = T_1$, the two gradients $\nabla_{z_{0|t}} G_1$ (for data consistency) and $\nabla_{z_{0|t}} G_2$ (for perceptual consistency) lie in the tangent space $\mathcal{T}_{z_0}\mathcal{Z}$. These gradients contribute to the update of the latent representation. The overall gradient update becomes:

$$z_{0|T_1}' = z_{0|T_1} - (\zeta_1 \nabla_{z_{0|T_1}} G_1 + \zeta_2 \nabla_{z_{0|T_1}} G_2) \tag{B-44}$$

Where $\zeta$ are scalars. Since both gradients reside in the tangent space $\mathcal{T}_{z_{0|T_1}}\mathcal{Z}$, and Assumption 1, the updated term $z_{0|T_1}'$ remains on the latent manifold $\mathcal{Z}$. Using the update rule for $z_{T_1-1}$, similar to the diffusion update step, we have:

$$z_{T_1-1} = \sqrt{\bar{\alpha}_{T_1-1}}z_{0|T_1}' + \sqrt{1 - \bar{\alpha}_{T_1-1}}\epsilon', \quad \epsilon' \sim \mathcal{N}(0, I) \tag{B-45}$$

Thus, the updated latent variable remains on the manifold, as the noise component $\epsilon'$ is Gaussian, and the mean update is based on $z_{0|T_1}' \in \mathcal{Z}$. Applying Lemma 2, give us $p(z_{T_1-1})$, that is probabilistically concentrated on $Z_{T-1}$.

The perceptual manifold, a subspace of the content-bias manifold defined by the data-consistency gradient $\nabla_{z_{0|t}} G_1$. In other words, data consistency keeps the sample within a region where the structural content is retained, and perceptual consistency term ensures that the sample moves toward regions of the manifold $\mathcal{Z}$ that are perceptually meaningful. The second gradient term $\nabla_{z_{0|t}} G_2(\psi_p(\mathcal{D}(z_{0|t})), \psi_p(y))$ represents a movement within this subspace to align with human perception. Since $\nabla_{z_{0|t}} G_2$ resides in the tangent space $\mathcal{T}_{z_0}\mathcal{Z}$ and is also influenced by the perceptual features $\psi_p$, the update ensures that the latent variable moves towards a more perceptually consistent locality within the overall content-bias region.

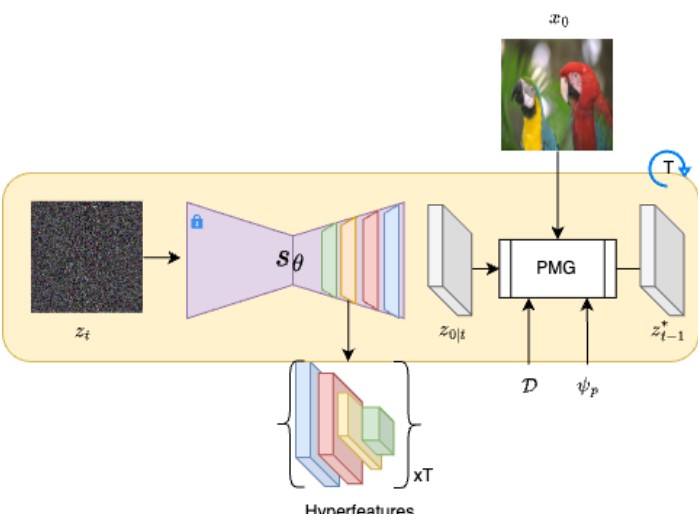

Figure 4: Illustrates PMG sampling and multi-scale and multi-timestep feature maps from denoising U-Net ($s_\theta$)
for NR-IQA. In the image, $z_t$ is intermediate noisy latent, $\mathcal{D}$ and $\psi$ are decoder and perceptual quality metric, respectively. PMG is our proposed algorithm for estimating the noisy sample $z_{t-1}^*$.

Formally, let $\mathcal{M}_{\text{content}}$ be the subspace of manifold corresponding to content consistency based on $G_1$, and let $\mathcal{M}_{\text{perceptual}} \subset \mathcal{M}_{\text{content}}$ be the sub-manifold that represents regions of perceptual consistency. The update using $\nabla_{z_{0|t}} G_2$ effectively ensures:

$$z''_{0|t} \in \mathcal{M}_{\text{perceptual}}, \tag{B-46}$$

where $\mathcal{M}_{\text{perceptual}}$ is a more constrained subspace within the content-consistent manifold, ensuring perceptual quality. By induction, we have shown that for all $t$, there exists a $z_0 \in \mathcal{Z}$ such that $z_t$ remains on the latent manifold throughout the diffusion process. Furthermore, the inclusion of the perceptual consistency gradient ensures that $z''_{0|t}$ is updated towards a perceptually consistent region on the manifold. Thus, the final updated latent variable $z''_{0|t}$ is not only data-consistent but also perceptually consistent within the manifold $\mathcal{Z}$, as required.

This concludes the proof.

## C IMPLEMENTATION DETAILS

### C.1 DATASETS AND EVALUATION PROTOCOL

The datasets used in our study (Table 1) contain images labeled with Mean Opinion Scores (MOS) following ITU-T P.910 guidelines ITU-T RECOMMENDATION (1999). We train the regressor $g_\phi$ using $l_2$ loss on MOS ground truth values. Evaluation metrics include Pearson Linear Correlation Coefficient (PLCC) and Spearman's Rank Order Correlation Coefficient (SRCC), ranging from 0 to 1, with higher values indicating better correlation.

Following Saha et al. (2023); Madhusudana et al. (2022), we split each dataset into training, validation, and test sets (70%, 10%, and 20%, respectively), using source image-based splits to prevent content overlap. The process is repeated 10 times, and median performance is reported to ensure robustness.

## C.2 Implementation Details

**Model Configuration**  For text conditioning, we use an empty string. We adopt the SDv1.5 and VQ-VAE from the official Stable Diffusion v1.5, with default settings from GitHub[1] and Hugging Face[2]. VQ-VAE is used with 8x downsampling for $512 \times 512$ resolution, which matches the typical resolution of IQA datasets like LIVE (Sheikh et al. (2006)) and CSIQ (Larson & Chandler (2010)).

**Sampling and Perceptual Features**  We use 10 DDIM steps for sampling, balancing efficiency and quality. The choice of perceptual metric $\psi_p$ is crucial—using well-correlated metrics such as RE-IQA or MUSIQ improves model performance. Poor metrics can degrade results, as shown in our ablation studies.

**Guidance Weights**  The weights for perceptual guidance, $\zeta_1$ and $\zeta_2$, are set to 1 and 0.2 based on empirical evaluations. This setup provides sufficient guidance, which enhances prediction quality.

## C.3 Autoencoder

Though a perfect autoencoder is ideal for maintaining samples on the manifold $\mathcal{M}$, the Stable Diffusion v1.5 VAE yields effective results despite minor imperfections. As shown in Table 6, it provides the best performance across configurations.

# D Additional Results and Ablation Study

In this section, we provide further empirical evaluations of our proposed Perceptual Consistency in Diffusion Models (PCDM) by presenting additional experimental results, ablation studies, and analyses to supplement the findings in the main paper. We also evaluate the impact of model hyperparameters and different configurations, including the version of Stable Diffusion (SD), the number of time steps, and the weighting of perceptual guidance terms. Lastly, we discuss the impact of various layers of UNet towards NR-IQA performance.

## D.1 Synthetic IQA Datasets

Table 5 reports the Pearson Linear Correlation Coefficient (PLCC) and Spearman Rank Order Correlation Coefficient (SRCC) scores of PCDM and existing NR-IQA methods on four synthetic datasets: LIVE (Sheikh et al. (2006)), CSIQ (Larson & Chandler (2010)), TID2013 (Ponomarenko et al. (2013)), and KADID (Lin et al. (2019)). The results demonstrate that our proposed approach outperforms other methods across almost all datasets. Similar to the performance on authentic distortion dataset in Table 2, PCDM-$\psi_{SDv1.5}$ achieves the best performance across all synthetic datasets, indicating a strong alignment with human perceptual judgments. The results also suggest that perceptual guidance using $\psi_{Re-IQA}$ and $\psi_{SDv1.5}$ consistently enhances the model's generalization capabilities. The superior performance of PCDM-$\psi_{SDv1.5}$ on the TID2013 (Ponomarenko et al. (2013))and KADID (Lin et al. (2019)) datasets, with PLCC and SRCC scores of 0.921/0.883 and 0.961/0.958, respectively, underscores the value of utilizing diffusion hyperfeatures.

## D.2 Impact of Time Step Range on Sampling Process

Figure 5 shows the effect of varying the range of time steps used during the sampling process. We observe a general decline in SRCC as we increase the time range, specifically when more noisy samples are involved. For larger time step ranges, the model relies on noisier intermediate representations, which reduces its ability to accurately predict image quality. This is an expected behaviour since, details in the images are generated towards the later time steps in diffusion process,i.e. less noise. This suggests that optimizing the range of time steps used for feature extraction is critical to maintaining high-quality predictions.

---

[1] https://github.com/CompVis/stable-diffusion
[2] https://huggingface.co/CompVis,
https://huggingface.co/stable-diffusion-v1-5/stable-diffusion-v1-5

| Methods | LIVE | | CSIQ | | TID2013 | | KADID | |
|---|---|---|---|---|---|---|---|---|
| | PLCC | SRCC | PLCC | SRCC | PLCC | SRCC | PLCC | SRCC |
| ILNIQE | 0.906 | 0.902 | 0.865 | 0.822 | 0.648 | 0.521 | 0.558 | 0.534 |
| BRISQUE | 0.944 | 0.929 | 0.748 | 0.812 | 0.571 | 0.626 | 0.567 | 0.528 |
| WaDIQaM | 0.955 | 0.960 | 0.844 | 0.852 | 0.855 | 0.835 | 0.752 | 0.739 |
| DBCNN | 0.971 | 0.968 | 0.959 | 0.946 | 0.865 | 0.816 | 0.856 | 0.851 |
| TIQA | 0.965 | 0.949 | 0.838 | 0.825 | 0.858 | 0.846 | 0.855 | 0.850 |
| MetaIQA | 0.959 | 0.960 | 0.908 | 0.899 | 0.868 | 0.856 | 0.775 | 0.762 |
| P2P-BM | 0.958 | 0.959 | 0.902 | 0.899 | 0.856 | 0.862 | 0.849 | 0.840 |
| HyperIQA | 0.966 | 0.962 | 0.942 | 0.923 | 0.858 | 0.840 | 0.845 | 0.852 |
| TReS | 0.968 | 0.969 | 0.942 | 0.922 | 0.883 | 0.863 | 0.859 | 0.859 |
| MUSIQ | 0.911 | 0.940 | 0.893 | 0.871 | 0.815 | 0.773 | 0.872 | 0.875 |
| RE-IQA | 0.971 | 0.970 | 0.960 | 0.947 | 0.861 | 0.804 | 0.885 | 0.872 |
| LoDA | 0.979 | 0.975 | - | - | 0.901 | 0.869 | 0.936 | 0.931 |
| PCDM-$\psi_\phi$ | 0.980 | 0.978 | 0.971 | 0.952 | 0.871 | 0.814 | 0.892 | 0.885 |
| PCDM-$\psi_{BRISQUE}$ | 0.980 | 0.977 | 0.970 | 0.951 | 0.814 | 0.771 | 0.841 | 0.838 |
| PCDM-$\psi_{MUSIQ}$ | 0.981 | 0.979 | 0.969 | 0.950 | 0.869 | 0.811 | 0.898 | 0.887 |
| PCDM-$\psi_{Re-IQA}$ | 0.983 | 0.981 | 0.972 | 0.952 | 0.904 | 0.882 | 0.932 | 0.930 |
| PCDM-just $\psi_{SDv1.5}$ | 0.981 | 0.978 | 0.971 | 0.958 | 0.908 | 0.876 | 0.935 | 0.931 |
| **PCDM-$\psi_{SDv1.5}$** | **0.988** | **0.986** | **0.981** | **0.964** | **0.921** | **0.883** | **0.961** | **0.958** |

Table 5: Comparison of our proposed PCDM with SOTA NR-IQA methods on PLCC and SRCC Scores for Synthetic IQA datasets. The best results are highlighted in bold.

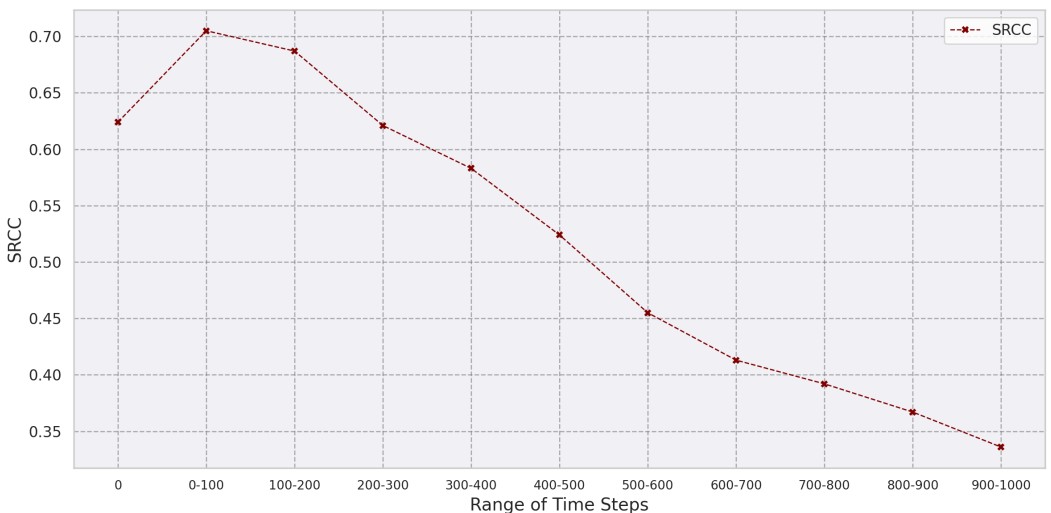

Figure 5: We report the behaviour of PCDM as we change the range of timesteps in the sampling process. As we move towards larger timestep range buckets, where there are more noisy samples, SRCC on FLIVE (Ying et al. (2020)) decreases.

### D.3 STABLE DIFFUSION MODEL VERSIONS

Table 6 evaluates the effect of different versions of Stable Diffusion (SD) on the CLIVE (Ghadiyaram & Bovik (2015)) and FLIVE (Ying et al. (2020)) datasets. The results indicate that SD v1.5 consistently outperforms other versions, achieving the highest PLCC and SRCC scores for both datasets. Specifically, SD v1.5 reaches an SRCC of 0.908 on CLIVE and 0.705 on FLIVE, outperforming newer versions like v2.0 and v2.1, which exhibit lower correlation values. The decline in performance in newer versions may be attributed to architectural changes or training modifications

| SD Version | CLIVE | | FLIVE | |
|---|---|---|---|---|
| | PLCC↑ | SRCC↑ | PLCC↑ | SRCC↑ |
| 1.3 | 0.932 | 0.901 | 0.804 | 0.695 |
| 1.4 | 0.938 | 0.903 | 0.807 | 0.698 |
| 1.5 | **0.940** | **0.908** | **0.812** | **0.705** |
| 2 | 0.910 | 0.882 | 0.781 | 0.674 |
| 2.1 | 0.917 | 0.886 | 0.788 | 0.681 |

Table 6: PLCC and SRCC Scores for Different Versions of SD on CLIVE (Ghadiyaram & Bovik (2015)) and FLIVE (Ying et al. (2020)). The best results are highlighted in bold, and the second-best results are underlined.

that diverge from the characteristics required for effective NR-IQA, i.e. focus on the generation of high quality aesthetic image, rather than a broader coverage of image quality.

| Time Steps | SRCC↑ | Time Taken (s) ↓ |
|---|---|---|
| 1 | 0.624 | 3.27 |
| 5 | 0.673 | 9.89 |
| 10 | 0.705 | 21.30 |
| 50 | 0.711 | 110.45 |

Table 7: SRCC scores and time taken for different timesteps on FLIVE (Ying et al. (2020)) by PCDM-$\psi_{SDv1.5}$.

## D.4    EFFECT OF TIME STEPS ON QUALITY AND COMPUTATION TIME

Table 7 provides an analysis of the SRCC scores and the computation time for different numbers of time steps on the FLIVE (Ying et al. (2020)) dataset using PCDM-$\psi_{SDv1.5}$. As expected, increasing the number of time steps improves the SRCC score, with the highest value of 0.711 obtained at 50 time steps. However, this comes at the cost of increased computational time, with a significant jump from 21.30 seconds for 10 time steps to 110.45 seconds for 50 time steps. This trade-off suggests that while more time steps can yield better performance, it is essential to balance quality with computational efficiency, especially for real-time applications.

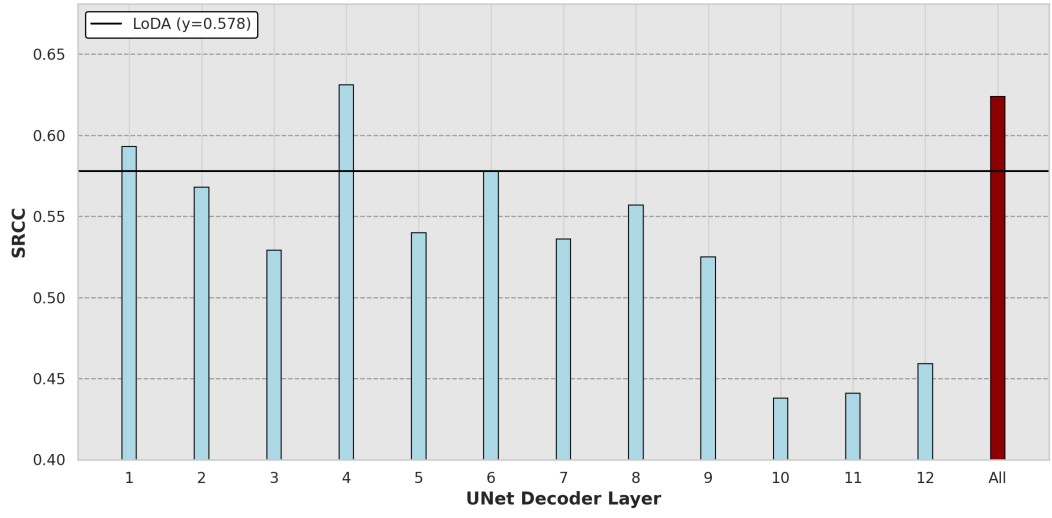

Figure 6: Contribution of individual layer of SDv1.5 towards SRCC for FLIVE (Ying et al. (2020)). For the sake of computational cost, we run single step sampling and extract individual layers and calculate SROCC.

## D.5 Contribution of Individual Layers

Figure 6 illustrates the contribution of individual layers of SD v1.5 towards SRCC on the FLIVE dataset. Using a single-step sampling, we analyze how different layers contribute to image quality predictions. The results indicate that intermediate layers, such as Layer 4, contribute significantly more to the final quality prediction than other layers. This layer-wise analysis is crucial in understanding which layers are more relevant for NR-IQA, providing insights for potential optimization by focusing on the most informative layers.

| $\zeta_2$ | CLIVE | | FLIVE | |
|---|---|---|---|---|
| | PLCC↑ | SRCC↑ | PLCC↑ | SRCC↑ |
| 0 | 0.853 | 0.842 | 0.751 | 0.672 |
| 0.2 | **0.940** | **0.908** | **0.812** | **0.705** |
| 0.5 | 0.931 | 0.901 | 0.802 | 0.691 |
| 0.7 | 0.918 | 0.882 | 0.780 | 0.675 |
| 1 | 0.904 | 0.868 | 0.772 | 0.664 |

Table 8: PLCC and SRCC Scores for varying the weights of second term in PMG (equation 13) on CLIVE (Ghadiyaram & Bovik (2015)) and FLIVE (Ying et al. (2020)). The best results are highlighted in bold, and the second-best results are underlined.

## D.6 Effect of Perceptual Guidance Weighting

Table 8 presents the PLCC and SRCC scores for varying the perceptual guidance weight, $\zeta_2$, in the Perceptual Manifold Guidance (PMG) term for the CLIVE (Ghadiyaram & Bovik (2015)) and FLIVE (Ying et al. (2020)) datasets. The results indicate that an optimal value of $\zeta_2 = 0.2$ yields the best PLCC and SRCC scores all ten datasets. Specifically, an SRCC of 0.908 is achieved on CLIVE (Ghadiyaram & Bovik (2015)) and 0.705 on FLIVE (Ying et al. (2020)) at this weight. When $\zeta_2$ is set too high (e.g., $\zeta_2 = 1$), the model's performance deteriorates, suggesting that samples move away from the perceptually consistent region on manifold. Conversely, setting $\zeta_2$ too low results in underutilization of the perceptual guidance, which leads to suboptimal quality predictions. With moderate weighting, superior performanc can be achieved across all benchmarks.

## D.7 Summary and Insights

Our extended experiments validate the effectiveness of PCDM across various IQA datasets, demonstrating its robustness against both synthetic and real-world distortions. The ablation studies provide valuable insights into the factors that impact model performance:

- Model Version Selection: SD v1.5 emerged as the best-performing version for NR-IQA, emphasizing the importance of selecting the appropriate diffusion model.

- Time Step Range: If sampling time steps from higher ranges are taken, it diminishes the performance significantly.

- Total Time steps: While increasing time steps improves prediction quality, it also significantly increases computation time, highlighting a trade-off between accuracy and efficiency.

- Layer Importance: Intermediate layers of the diffusion model were found to contribute the most towards perceptual quality, suggesting the possibility of optimizing feature extraction by focusing on specific layers.

- Perceptual Guidance Weighting: The optimal perceptual guidance weight strikes a balance between content and perceptual terms, which is crucial for maintaining high-quality predictions.

Overall, these results underscore the capability of pretrained diffusion models to serve as effective feature extractors for NR-IQA tasks, provided that the appropriates guidance is provided. Our method, which exploits the inherent generalization capabilities of diffusion models, successfully

advances the state-of-the-art in NR-IQA, offering a promising approach for future developments in perceptually consistent no-reference image quality assessment.

# E  LIMITATIONS & EXTENSION

Our proposed PCDM framework shows strong performance in NR-IQA; however, there are a few limitations and potential extensions worth noting.

**Limitations:** The computational cost of our approach, particularly due to the iterative diffusion model sampling, can be high, which might limit real-time or resource-constrained applications. Also, the scope of our work is limited to NR-IQA. We did not extend our evaluation to other low-level vision tasks or explore the use of perceptual control in image generation due to time constraints and the focus on a single task in this paper.

**Extensions:** The general framework of PCDM, particularly its ability to extract perceptual features, has potential applications beyond NR-IQA. It can be extended to other low-level vision tasks that are sensitive to perceptual features, such as image denoising, super-resolution, and enhancement, where quality assessment is crucial. Moreover, since posterior sampling is known to be limiting due to its clean sample estimation from intermediate time step $t$, one can explore using more advanced techniques such as Bayesian filtering to directly approximate posterior sampling in intermediate time step $t$, that is computationally more efficient as it avoids taking derivatives on the estimated score function. Our method can also be adapted for perceptually controllable image generation. Exploring these directions can help expand the impact of our approach and leverage its strengths across a broader range of vision tasks.

In future work, we plan to explore these extensions, allowing PCDM to contribute more broadly to the field of low-level computer vision.

