# OpenReview forum: "PCDM: PERCEPTUAL CONSISTENCY IN DIFFUSION MODELS FOR NO-REFERENCE IMAGE QUALITY ASSESSMENT"
_ICLR.cc/2025/Conference — ICLR 2025 Conference Withdrawn Submission_

### Official Review · Reviewer_LCN8 · 2024-10-20

**Soundness:** 1
**Presentation:** 2
**Contribution:** 1
**Rating:** 1
**Confidence:** 5

**Summary:**

This work presents an NR-IQA model based on a latent diffusion model. It employs a multi-scale and multi-time feature extraction strategy.

**Strengths:**

+ Using diffusion model for NR-IQA is interesting.

**Weaknesses:**

- This work claims that it address the limited applicability of pixel diffusion models by using the more efficient LDMs. However, there is no comparison between pixel diffusion model-based and LDM-based solutions.
- There is no specific definion of perceptually consistent region. Neither, it is unclear why the proposed PGM is beneficial for NR-IQA.
- The paper is very difficult to follow. After reading the paper, the Reviewer still can't understand while diffusion models generative models, why and how they can be used for NR-IQA, a discriminative task?
- The multi-scale feature extraction is not novel, which has been used in many prior arts.
- The proposed method is claimed as a zero-shot NR-IQA. However, it still need to be trained on IQA datasets as described in L358-L364.
- Given the above concerns, the reported superior results is not convincing.

**Questions:**

See the weaknesses.

---

### Official Review · Reviewer_v1ZK · 2024-10-29

**Soundness:** 3
**Presentation:** 2
**Contribution:** 2
**Rating:** 5
**Confidence:** 4

**Summary:**

The author proposed an algorithm called Perceptual Manifold Guidance (PMG), which utilizes a pre-trained latent diffusion model and a perceptual quality metric to extract perceptually consistent multi-scale and multi-temporal step features from a denoising U-Net, referred to as diffusion hyper-features. The method named Perceptual Consistency in Diffusion Model (PCDM) claims to be the first to explore perceptual consistency in diffusion models for zero-shot NR-IQA.

**Strengths:**

1. The paper proposes a pioneering method for zero-shot NR-IQA using a latent diffusion model, which significantly differs from traditional methods that rely on reference images or large training datasets.
2. Experiments conducted on multiple IQA datasets demonstrate the effectiveness of the method.
3. The authors provide a solid theoretical foundation for their proposed algorithm, including detailed proofs of the approach.

**Weaknesses:**

1. The manuscript lacks comparisons with the latest methods, such as LIQE[1] and DEIQT[2].
2. Does this method heavily rely on the availability and quality of pre-trained latent diffusion models, limiting its applicability?
3. Although there are many theoretical proofs, there is no corresponding visualization to demonstrate the effectiveness of the algorithm. Visualizations of the manuscript's method for extracting features from distorted regions should be added.
4. Ablation experiments were only demonstrated on the fLivE dataset, lacking ablation effects of different components on more datasets.
5. There is a lack of comparison regarding the number of model parameters.
6. The method presented in the manuscript performs better on synthetic datasets than on real ones; could it be considered overfitting to specific datasets? The authors are requested to discuss in depth the reasons for this performance discrepancy and provide visualizations from different domain datasets to substantiate this reason.
7. Please include more experiments in cross-dataset validation with real datasets and synthetic datasets, such as training on Koniq or CLIVE and testing on FLIVE, as well as training on Koniq (Kadid) and testing on Kadid (Koniq).


[1] Zhang W, Zhai G, Wei Y, et al. Blind image quality assessment via vision-language correspondence: A multitask learning perspective[C]//Proceedings of the IEEE/CVF conference on computer vision and pattern recognition. 2023: 14071-14081.
[2] Qin G, Hu R, Liu Y, et al. Data-efficient image quality assessment with attention-panel decoder[C]//Proceedings of the AAAI Conference on Artificial Intelligence. 2023, 37(2): 2091-2100.

**Questions:**

1. The authors claim their work on zero-shot, but is there a lack of auxiliary validation on few-shot?
2. Why are only three methods listed in Table 3?
3. In cross-dataset validation, why are there no experiments conducted on the FLIVE dataset?
4. Is there an optimal subset of diffusion hyper-features that can maximize performance, or is the strategy of aggregating all features the best strategy?
5. How scalable is this method with increasing image resolution or more complex image datasets, and what are the computational trade-offs involved?

---

### Official Review · Reviewer_cxiP · 2024-11-04

**Soundness:** 3
**Presentation:** 3
**Contribution:** 2
**Rating:** 6
**Confidence:** 3

**Summary:**

This paper introduced the Perceptually Consistent Diffusion Model (PCDM) for No-Reference Image Quality Assessment (NR-IQA). By leveraging the powerful representation capabilities of pre-trained latent diffusion models (LDMs), they proposed Perceptual Manifold Guidance (PMG) to direct the sampling process toward perceptually consistent regions on the data manifold. The experiments are quite promising.

**Strengths:**

The greatest strength of this paper is that it can effectively extract multi-scale features from LDMs for the NRIQA task. Although it is not highly complex, the experimental results are excellent, which fully validates the advanced nature of the proposed method.

**Weaknesses:**

As I am not an expert in the field of Image Quality Assessment (IQA), I find myself focusing more on the algorithmic details. However, I noticed that the paper lacks certain explanations.

1. For instance, the structure of the lightweight regression network  g_phi mentioned in Equation (16) is not clearly defined. It would be helpful to know the number of parameters it includes and how different structures might affect the experimental outcomes. I recommend providing supplementary details on this.

2. Additionally, the function psi_p  in Equation (13) is not clearly explained. Could you please elaborate on its role and significance?

3. Furthermore, more detailed explanations for Figure 1 would be beneficial. Specifically, why does it align with human visual perception? Clarifying this would strengthen the paper.

4. To enhance persuasiveness, it would be better to conduct a more comprehensive analysis comparing the proposed method with existing No-Reference IQA (NRIQA) methods that utilize diffusion models. Such an analysis would further demonstrate the effectiveness of your proposed algorithm.

**Questions:**

Please see the Weaknesses part.

---

### Official Review · Reviewer_5zTh · 2024-11-05

**Soundness:** 2
**Presentation:** 2
**Contribution:** 2
**Rating:** 5
**Confidence:** 5

**Summary:**

The main takeaway from this paper is the authors' claim that the intermediate features extracted from the diffusion model provide a strong representation for NR-IQA tasks. In other words, the diffusion model acts as an effective feature extractor for IQA. The method draws inspiration from the readout and hyper-feature networks, where noise is added to clean latent features to extract intermediate multi-scale features from the U-Net. These features are then used to predict specific properties, such as depth or something else. In this context, author use those feature to predict IQA value.

**Strengths:**

(+) The idea of using a diffusion model as a feature extractor for IQA tasks is an interesting exploration.

**Weaknesses:**

(-) My first concern is the key source of the performance of the proposed IQA method. Does the improvement truly stem from the multi-scale features within the U-Net's prior knowledge? The use of intermediate features in tasks like depth estimation, correspondence, and human pose in hyperfeature network [1] or readout [2] is justified because guidance needs to occur during denoising stages. But for IQA, why would multi-scale features extracted from noisy latent representations be beneficial for quality assessment? Could the authors provide a justification? Figure 5 illustrates this concern: the correlation improves significantly at smaller time steps, where the features are derived from cleaner latent representations. This suggests that less noisy features naturally yield better quality estimates, which means noisy latent feature seems have no benefit for IQA task. Therefore, returning to my initial question: what is the true source of the method’s performance? If the cleanest features lead to optimal performance, why doesn’t the method simply use near noise-free features from the really small time step, as hinted at in line 1290? In that context, the main feature extractor is the VAE, which seems more reasonable to be a good feature extractor for IQA task.

(-) Could the authors explain why peak IQA performance is achieved when some noise remains time steps from [0-100]?

(-) The approach, as shown in Figure 4 of the appendix, appears to be inspired by series work of readout method [2] and hyper-feature network [1], but the citation and discussion for readout is missing.

(-) Regarding Figure 5, what does the range [0–100] signify? Does it represent the mean SRCC performance across these time steps? Clarifying this would improve the explanation.

(-) The authors claim in Figure 1 that their method enables guidance and transitions on the latent manifold. From my understanding, this transition refers to improving image quality by optimizing noisy latent features, similar to what the readout method does. Is that correct? Specifically, training a small network to predict IQA scores from latent features should allow these features (input to the U-Net) to be optimized for better image quality. If my understanding is accurate, I couldn't find any experimental evidence in the paper to support this claim about the transition.

(-) My second concern is the potential degradation caused when mapping images from RGB space to latent space using the VAE in stable diffusion. This process can sometimes result in color tone shifts or a loss of texture details, adding further degradation to images that are already compromised. Could the authors address this issue and explain its impact on their IQA evaluations?

(-) I’m confused about the choice of the perceptual metric. As I understand it, using BRISQUE implies that the predicted IQA values are optimized to match the BRISQUE scores. If this understanding is correct, what does it mean when the chosen perceptual metric is referred to as "sdv1.5" or "just-sdv1.5"? What IQA metric is being used as the label in that case?

(-) It would be helpful if the authors could include an appendix summarizing all the mathematical formulas on a single page.

[1] Luo, Grace, et al. "Diffusion hyperfeatures: Searching through time and space for semantic correspondence." Advances in Neural Information Processing Systems 36 (2024).
[2] Luo, Grace, et al. "Readout guidance: Learning control from diffusion features." Proceedings of the IEEE/CVF Conference on Computer Vision and Pattern Recognition. 2024.

**Questions:**

All my questions are outlined in the weaknesses section. The authors assume the encoder and decoder are perfect, but the diffusion model components introduce significant extra degradation. Specifically, the original SDXL can add distortions, such as those caused by the VAE, and the hyper-feature network or readout network operates on noisy latent features, contributing further degradation. I am curious how the authors justify performing accurate image quality assessment on already degraded images while introducing even more degradation. Additionally, the paper's presentation could be improved. Concepts related to the readout or hyper-feature network, which are relatively straightforward, should be illustrated with clear figures rather than using pseudo algorithms or text. For example, an pipeline figure highly illustrate what components are trainable during training or inference phase. It is wrapping simple concepts in complex stuffs.

---

### Note · Authors · 2024-11-15

I have read and agree with the venue's withdrawal policy on behalf of myself and my co-authors.